# TGR5 signalling promotes mitochondrial fission and beige remodelling of white adipose tissue

Laura A. Velazquez-Villegas[1], Alessia Perino[1], Vera Lemos[1,2], Marika Zietak[1,3], Mitsunori Nomura[1], Thijs Willem Hendrik Pols[1] & Kristina Schoonjans[1]

Remodelling of energy storing white fat into energy expending beige fat could be a promising strategy to reduce adiposity. Here, we show that the bile acid-responsive membrane receptor TGR5 mediates beiging of the subcutaneous white adipose tissue (scWAT) under multiple environmental cues including cold exposure and prolonged high-fat diet feeding. Moreover, administration of TGR5-selective bile acid mimetics to thermoneutral housed mice leads to the appearance of beige adipocyte markers and increases mitochondrial content in the scWAT of $Tgr5^{+/+}$ mice but not in their $Tgr5^{-/-}$ littermates. This phenotype is recapitulated in vitro in differentiated adipocytes, in which TGR5 activation increases free fatty acid availability through lipolysis, hence fuelling β-oxidation and thermogenic activity. TGR5 signalling also induces mitochondrial fission through the ERK/DRP1 pathway, further improving mitochondrial respiration. Taken together, these data identify TGR5 as a druggable target to promote beiging with potential applications in the management of metabolic disorders.

---

[1] Institute of Bioengineering, Ecole Polytechnique Fédérale de Lausanne, CH-1015 Lausanne, Switzerland. [2] Abel Salazar Biomedical Sciences Institute, University of Porto, 4050-013 Porto, Portugal. [3] Institute of Animal Reproduction and Food Research, Polish Academy of Sciences, 10-748 Olsztyn, Poland. Laura A. Velazquez-Villegas and Alessia Perino contributed equally to this work. Correspondence and requests for materials should be addressed to K.S. (email: kristina.schoonjans@epfl.ch)

Obesity and its associated metabolic comorbidities are not only posing an important medical problem but also have a significant societal impact. Novel preventive and therapeutic strategies to limit and/or reverse obesity are therefore receiving increasing attention from the biomedical research community. Recently, brown-like cells, also referred to as beige or brite cells, have been described within the white adipose tissue (WAT)[1,2]. Such beige cells display functional characteristics similar to brown adipocytes, including the ability to convert chemical energy into heat via the expression of uncoupling protein 1 (UCP1), known to dissipate the mitochondrial electrochemical gradient through proton leak and to induce uncoupled respiration[3,4]. Through this process, beiging of WAT increases the utilisation of nutrients and may contribute to whole-body energy homeostasis[1,5–7].

In coherence with their functional similarities, beige and brown cell differentiation also involves a shared transcriptional cascade that mainly relies on the transcription factors peroxisome proliferator-activated receptor gamma (PPARγ) and CCAAT/enhancer-binding proteins (C/EBPs)[8,9]. Additional transcriptional regulators that orchestrate brown and beige adipose-related gene networks include the PR domain-containing protein-16 (PRDM16)[10–12], that confers brown fat cell identity by promoting brown fat-specific gene expression through co-regulation of C/EBPs and PPARγ[10–12] and the peroxisome proliferator-activated receptor-coactivator-1 alpha (PGC-1α), which co-activates PPARγ and fine-tunes mitochondrial biogenesis, oxidative metabolism and thermogenesis[13].

Cold exposure is one of the main triggers of adipose tissue beiging. Reduction in ambient temperature elicits the release of

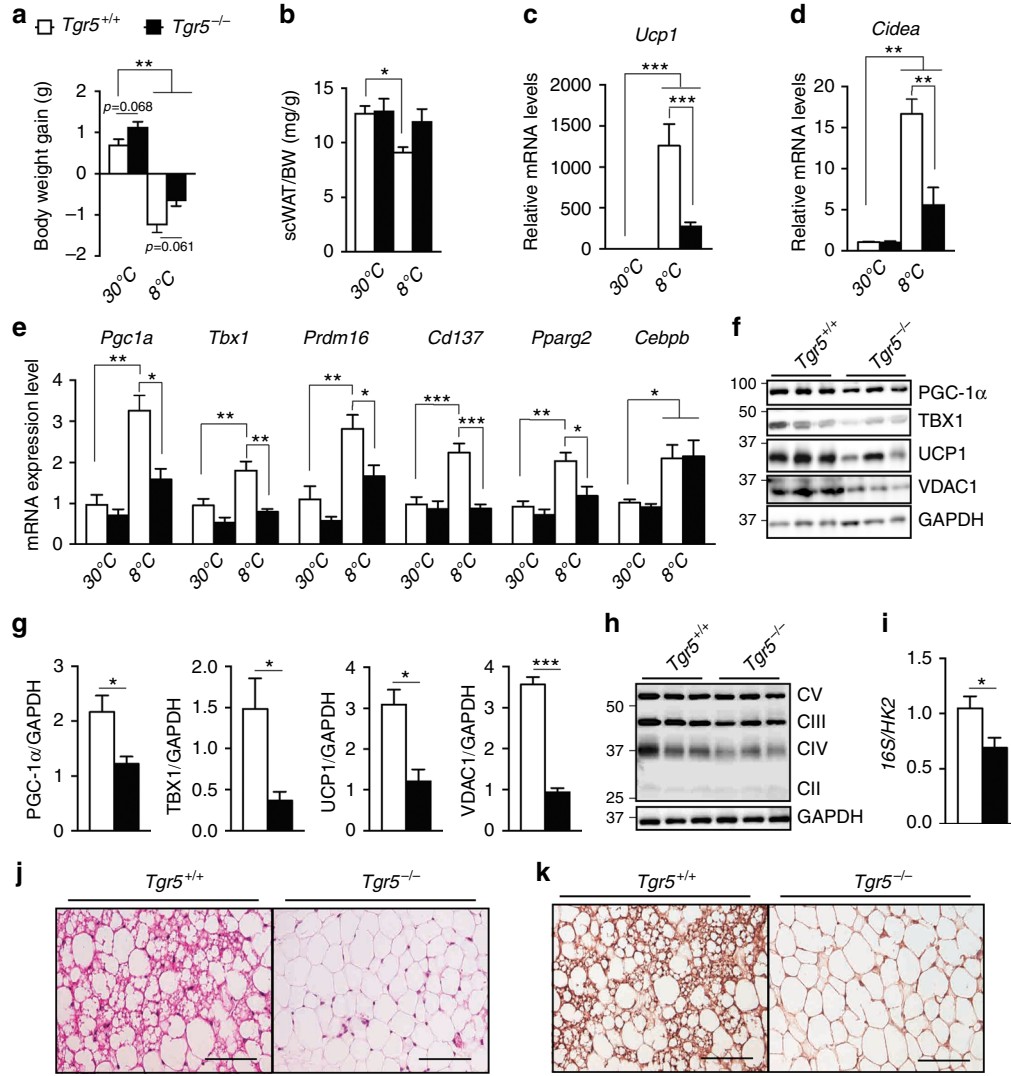

**Fig. 1** TGR5 is required for cold-induced scWAT beiging. **a** Body weight gain of TGR5 wild-type (*Tgr5^{+/+}*) and germline TGR5 knockout (*Tgr5^{−/−}*) mice housed at thermoneutrality (30 °C) or exposed to cold (8 °C) for 7 days. *n* = 10 per group. **b** scWAT over body weight (BW) ratio of mice described in **a**. **c–e** mRNA levels of beige remodelling markers *Ucp1* (**c**), *Cidea* (**d**), *Pgc1a, Tbx1, Prdm16, Cd137, Pparg2* and *Cebpb* (**e**) in the scWAT of mice described in **a**. **f** Representative (*n* = 10 per group) western blot of PGC-1α, the mitochondrial marker VDAC1 and beiging markers TBX1 and UCP1 from the scWAT of cold-exposed mice described in **a**. GAPDH was used as loading control. **g** Quantitative densitometry of the western blots showed in **f**. **h** Representative (*n* = 10 per group) western blot of mitochondrial OXPHOS complexes (CII–CV) from the scWAT of cold-exposed mice described in **a**, GAPDH was used as loading control. **i** Quantification of mitochondrial (16S) vs. nuclear (HK2) DNA ratio from the scWAT of cold-exposed mice described in **a**. **j, k** Representative haematoxylin and eosin (**j**) and UCP1 (**k**) stainings of scWAT sections from cold-exposed mice described in **a**. Scale bars = 50 μm. Results represent mean ± SEM. *P ≤ 0.05, **P ≤ 0.01 and ***P ≤ 0.001 vs. *Tgr5^{+/+}* group (at 30 and/or 8 °C) by one-way ANOVA followed by Bonferroni post hoc test (**a–e**) or Student's *t*-test (**g, i**). Uncropped western blots are provided in Supplementary Fig. 9A–C

norepinephrine from sympathetic fibres[14,15], and promotes beige fat differentiation and thermogenic capacity through β3-adrenergic receptor (β3-AR) signalling[14,16]. Interestingly, cold exposure also induces changes in bile acid (BA) composition, including an increase in secondary BAs[17,18].

BAs, beyond their classical role in dietary lipid absorption, are signalling molecules and exert significant systemic metabolic effects, including the promotion of energy expenditure in the brown adipose tissue (BAT) and muscle[19]. BA signalling is mainly mediated by the nuclear farnesoid X receptor (FXR) and the membrane G protein-coupled receptor TGR5[20–22]. TGR5 has a broad expression pattern and is responsible for many of the systemic metabolic actions of BAs including protection against diet-induced obesity[19,23,24], insulin resistance[25,26], liver steatosis[26] and atherosclerosis[27]. Dietary supplementation of TGR5-selective BA mimetics, such as 6-ethyl-23(S)methyl-cholic acid (EMCA, INT-777), efficiently prevents body weight gain in mice on a high-fat (HF) diet[19,26]. Although it has been demonstrated that TGR5 activation increases energy expenditure and fat mass oxidation by stimulating the basal metabolic rate[19,28], it is unknown whether beige cell formation within white fat depots contributes to this phenotype.

Here, we identify the BA–TGR5 axis as a novel molecular circuitry to enhance subcutaneous WAT (scWAT) beiging after cold exposure or HF diet feeding. More importantly, we demonstrate that TGR5 activation is sufficient to induce scWAT beiging at thermoneutrality, a condition in which the sympathetic drive to the fat is negligible. Furthermore, we show that TGR5-specific BA mimetics increase lipolysis and modulate the mitochondrial number and morphology in mature white adipocytes. The enhanced release of free fatty acids (FFAs) not only fuels β-oxidation, but also promotes UCP1-dependent thermogenesis and mitochondrial fission, the latter process being entirely dependent on the presence of an intact ERK1–DRP1 signalling axis. These results confirm the role of TGR5 as a target to modulate mitochondrial function and beige remodelling in the adipose tissue.

## Results

**TGR5 is required for cold-induced beiging of scWAT.** Cold exposure is one of the most potent physiological inducers of WAT beiging[29,30]. Moreover, secondary BAs—potent endogenous ligands of TGR5— are elevated in plasma of mice maintained at reduced ambient temperature[18]. To study the importance of the BA–TGR5 axis in cold-induced beiging of scWAT, we used TGR5 wild-type ($Tgr5^{+/+}$) and TGR5 germline knock-out ($Tgr5^{-/-}$) mice and exposed them to an ambient temperature of 8 °C for 1 week with free access to food and water. $Tgr5^{+/+}$ and $Tgr5^{-/-}$ mice at thermoneutrality (30 °C) were used as proper controls for this study. As expected, there was a significant difference in body weight between mice at 30 and 8 °C (Fig. 1a). If we compare the genotypes, $Tgr5^{-/-}$ mice presented a trend towards more body weight gain at 30 °C ($P = 0.068$) and less body weight loss ($P = 0.061$) after cold exposure (Fig. 1a). In addition, subcutaneous fat mass was significantly reduced in $Tgr5^{+/+}$ mice after cold exposure when compared to mice at thermoneutrality (Fig. 1b). Notably, $Tgr5^{+/+}$, but not their $Tgr5^{-/-}$ littermates, developed beiging in the scWAT after 1 week of reduced ambient temperature, as evidenced by the increase in messenger RNA (mRNA) levels of multiple brown fat-specific genes including *Ucp1* (Fig. 1c), the lipid droplet-associated protein cell death-inducing DFFA-like effector A (*Cidea*) (Fig. 1d), *Pgc1a*, *Prdm16*, *Pparg2* and specific beige adipocyte markers, including the transcription factor T-box 1 (*Tbx1*) and the tumour necrosis factor receptor superfamily, member 9 (*Tnfrsf9* or *Cd137*)

(Fig. 1e). Unlike the overall induction profile in $Tgr5^{+/+}$ mice, many of these genes remained unchanged between thermoneutral and cold exposed $Tgr5^{-/-}$ mice (Fig. 1c–e), supporting the notion that TGR5 is essential for scWAT beiging. Given the absence of any TGR5-dependent effect on the beiging markers under thermoneutral conditions, we focused our analysis on mice maintained at 8 °C. Compared to $Tgr5^{+/+}$ mice, a significant decrease of PGC-1α, TBX1, UCP1, the voltage-dependent anion channel 1 (VDAC1), a mitochondrial outer membrane marker, and mitochondrial complexes III and IV, was observed in the scWAT of $Tgr5^{-/-}$ mice (Fig. 1f–h; Supplementary Fig. 1A). Moreover, the mitochondrial over nuclear DNA ratio (mtDNA/nDNA—16S/HK2) was significantly reduced in scWAT from $Tgr5^{-/-}$ mice (Fig. 1i), suggesting that TGR5 is implicated in the regulation of mitochondrial biogenesis. Cold exposure induced the appearance of brown-like cells specifically within the scWAT of $Tgr5^{+/+}$ mice (Fig. 1j) and the beiging phenotype of those adipocytes was further confirmed by immunohistochemical detection of UCP1 (Fig. 1k). Although the expression of some genes in the epididymal WAT (epiWAT) and BAT was significantly changed (Supplementary Fig. 1B, C), we could not detect any difference at the protein or histological level between $Tgr5^{+/+}$ and $Tgr5^{-/-}$ mice (Supplementary Fig. 1D–J). Moreover, the expression values of all the beiging markers in the epiWAT were significantly lower than those found in scWAT and BAT, and this was also reflected at the protein level, confirming the specific role of TGR5 in selectively mediating subcutaneous beiging. Together, these results indicate that 1 week of cold exposure is sufficient to induce beiging of the scWAT and that this process is largely dependent on TGR5.

**Fat-specific deletion of TGR5 impairs cold-induced beiging.** To test the impact of TGR5 in adipocytes on the induction of scWAT beiging, we generated an inducible adipose tissue-specific $Tgr5^{-/-}$ model (called afterwards $Tgr5^{Adipoq-/-}$) by breeding mice carrying floxed $Tgr5$ alleles ($Tgr5^{fl/fl}$ mice) with the tamoxifen-inducible adiponectin ($Adipoq$)CreER$^{T2}$ mouse line. Validation of the $Tgr5^{Adipoq-/-}$ mouse line was carried out to verify the selective deletion of TGR5 in the adipose depots. The $Tgr5$ gene was indeed deleted in the scWAT and epiWAT and, to a lesser extent, in the BAT of the $Tgr5^{Adipoq-/-}$ mice compared to their controls (Supplementary Fig. 2A) while its expression was not reduced in other TGR5 expressing tissues (Supplementary Fig. 2B, C). To evaluate the importance of adipocyte TGR5 in mediating WAT beiging, we subjected $Tgr5^{Adipoq-/-}$ mice and their controls to the same cold challenge described in Fig. 1. $Tgr5^{Adipoq-/-}$ mice showed a significant reduction of their body temperature already after 48 h of cold exposure (Fig. 2a). This effect was maintained until the end of the experiment, showing that selective deletion of TGR5 in the adipose fat pads impairs the thermoregulation of these mice upon cold exposure. Similar to $Tgr5^{-/-}$ animals, mice with a targeted deletion of $Tgr5$ in adipocytes were not able to develop brown-like adipocytes (Fig. 2b), and to upregulate genes of the beige adipocyte signature in scWAT after cold exposure (Fig. 2c–e), with expression values of most of these genes comparable to those found in mice at 30 °C. $Tgr5^{Adipoq-/-}$ mice at 8 °C also displayed a marked reduction of beige markers at the protein level (Fig. 2f, g). The importance of TGR5 in the induction of beiging was also confirmed by the robust increase in UCP1 immunostaining of cold exposed $Tgr5^{Adipoq+/+}$, but not $Tgr5^{Adipoq-/-}$ scWAT (Fig. 2h). Moreover, the mitochondrial protein VDAC1 and the mtDNA/nDNA ratio were also significantly decreased in $Tgr5^{Adipoq-/-}$ mice compared to their controls (Fig. 2f, g, i). No significant changes in gene or protein expression (Supplementary Fig. 3A–F), or tissue

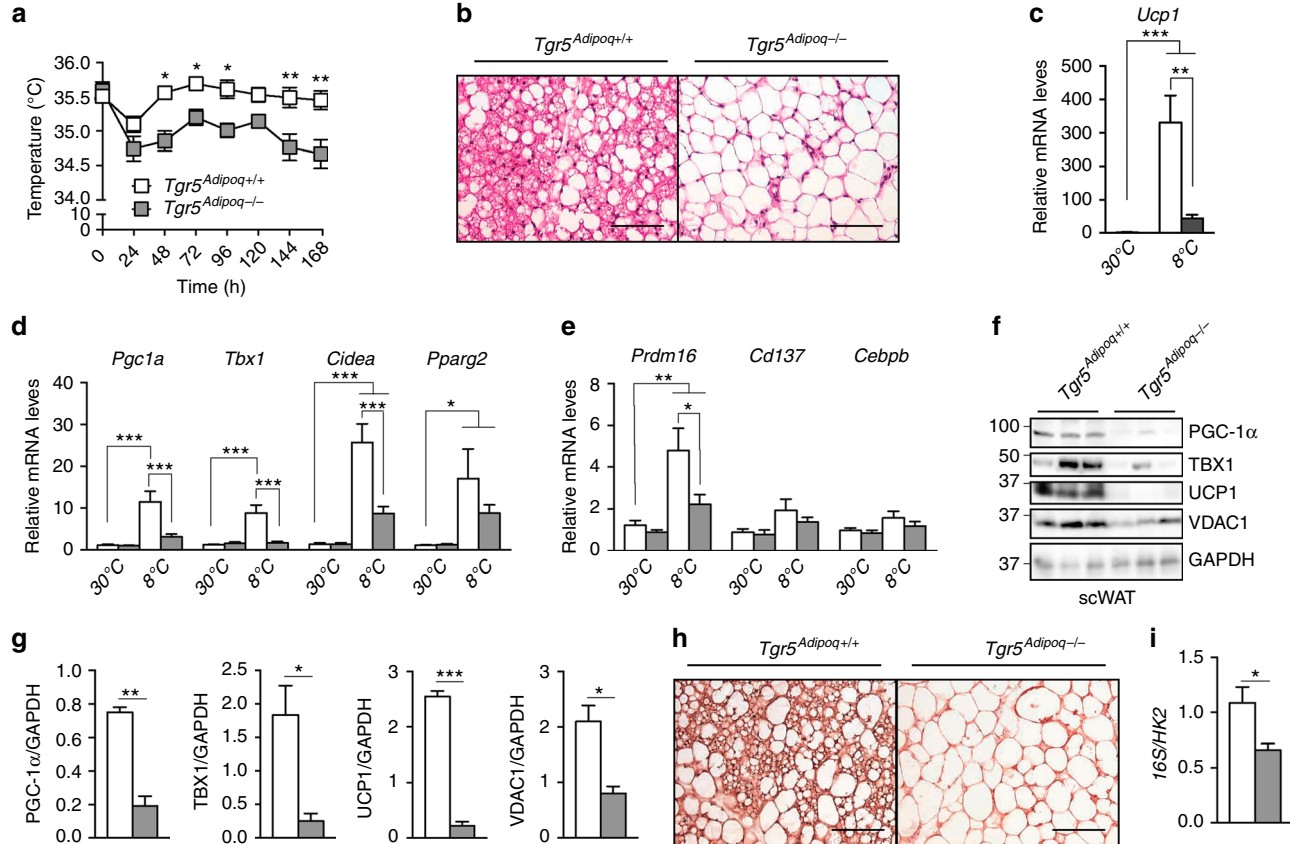

**Fig. 2** Adipocyte TGR5 is required for cold-induced scWAT beiging. **a** Body temperature of control mice (*Tgr5*$^{Adipoq+/+}$) and WAT-specific TGR5 knockout (*Tgr5*$^{Adipoq-/-}$) mice exposed to cold (8 °C) for 7 days. $n = 10$ per group. **b** Representative ($n = 5$ per group) haematoxylin and eosin stainings of scWAT of mice described in **a**. **c**–**e** mRNA levels of beige remodelling markers *Ucp1* (**c**) *Pgc1a, Tbx1, Cidea* and *Pparg2* (**d**), and *Prdm16, Cd137* and *Cebpb* (**e**) in the scWAT of *Tgr5*$^{Adipoq+/+}$ and *Tgr5*$^{Adipoq-/-}$ mice housed at thermoneutrality (30 °C) or exposed to cold (8 °C) for 7 days. $n = 10$ per group. **f** Representative ($n = 10$ per group) western blot of PGC-1α, the mitochondrial marker VDAC1 and beiging markers TBX1 and UCP1 from the scWAT of mice described in **a**. GAPDH was used as loading control. **g** Quantitative densitometry of the western blots showed in **f**. **h** Representative ($n = 5$ per group) UCP1 staining of scWAT sections from mice described in **a**. Scale bars = 50 μm. **i** Quantification of mitochondrial (16S) vs. nuclear (HK2) DNA ratio from the scWAT of mice described in **a**. Results represent mean ± SEM. *$P \leq 0.05$, **$P \leq 0.01$ and ***$P \leq 0.001$ vs. *Tgr5*$^{Adipoq+/+}$ group (at 30 and/or 8 °C) by one-way ANOVA (**c**–**e**) and two-way ANOVA (**a**) followed by Bonferroni post hoc test or Student's *t* test (**g**, **i**). Uncropped western blots are provided in Supplementary Fig. 10A–E

morphology (Supplementary Fig. 3G, H) were observed in the epiWAT and BAT of these mice. These results suggest that the beiging of scWAT after long-term cold exposure is strongly impaired after selective deletion of TGR5 in the adipocytes.

**TGR5 activation induces beiging of scWAT at thermoneutrality.** In order to evaluate a potential participation of the sympathetic nervous system in the induction of TGR5-dependent beige remodelling, *Tgr5*$^{+/+}$ and *Tgr5*$^{-/-}$ mice were maintained at thermoneutrality (30 °C) and subjected to a daily administration of the TGR5 selective BA mimetic INT-777 or vehicle for 1 week. Interestingly, activation of TGR5 significantly reduced body weight gain (Fig. 3a) and scWAT mass (Fig. 3b) and robustly induced expression of scWAT beiging markers, especially *Ucp1* and *Cidea* (Fig. 3c, d), specifically in *Tgr5*$^{+/+}$ mice. No changes in the expression of the adrenergic receptor beta 3 (*Adrb3*) and the rate-limiting enzyme catalysing the synthesis of catecholamines, tyrosine hydroxylase (*Th*) were noted (Fig. 3d), excluding compensatory effects from the adrenergic system. Selective TGR5 activation also increased the beige adipocyte signature at the protein level (Fig. 3e; Supplementary Fig. 4A) and induced the appearance of brown-like UCP1-positive adipocytes as demonstrated by immunostaining (Fig. 3f). These results demonstrate

that TGR5 activation is sufficient to induce the beige remodelling program in the scWAT, independent of environmental cues known to activate the sympathetic nervous system.

**TGR5 induces scWAT beiging after high-caloric intake.** Previous studies from our group demonstrated that supplementation of BAs efficiently prevents diet-induced obesity in mice on a HF diet[19]. These effects were partially explained by an increase in energy expenditure and fat oxidation in the BAT[19]. However, given the significant weight loss associated with BA administration and the recent discovery that chronic HF diet feeding induces the expression of beiging markers in the WAT[31], we explored the possibility that part of the changes in WAT mass and function could be explained by a TGR5-dependent induction of WAT-resident brown-like adipocytes. For this purpose, we used *Tgr5*$^{+/+}$ mice and fed them a HF diet for 20 weeks with or without INT-777 supplementation. In line with previous studies[19,26], INT-777 admixed to the HF diet blunted diet-induced body weight gain (Fig. 4a) and also triggered the appearance of beige adipocyte markers in the scWAT at the transcript (Fig. 4b) and protein (Fig. 4c; Supplementary Fig. 4B) levels. This effect again coincided with an increased number of mitochondria as evidenced by the augmentation of VDAC1 protein levels (Fig. 4c;

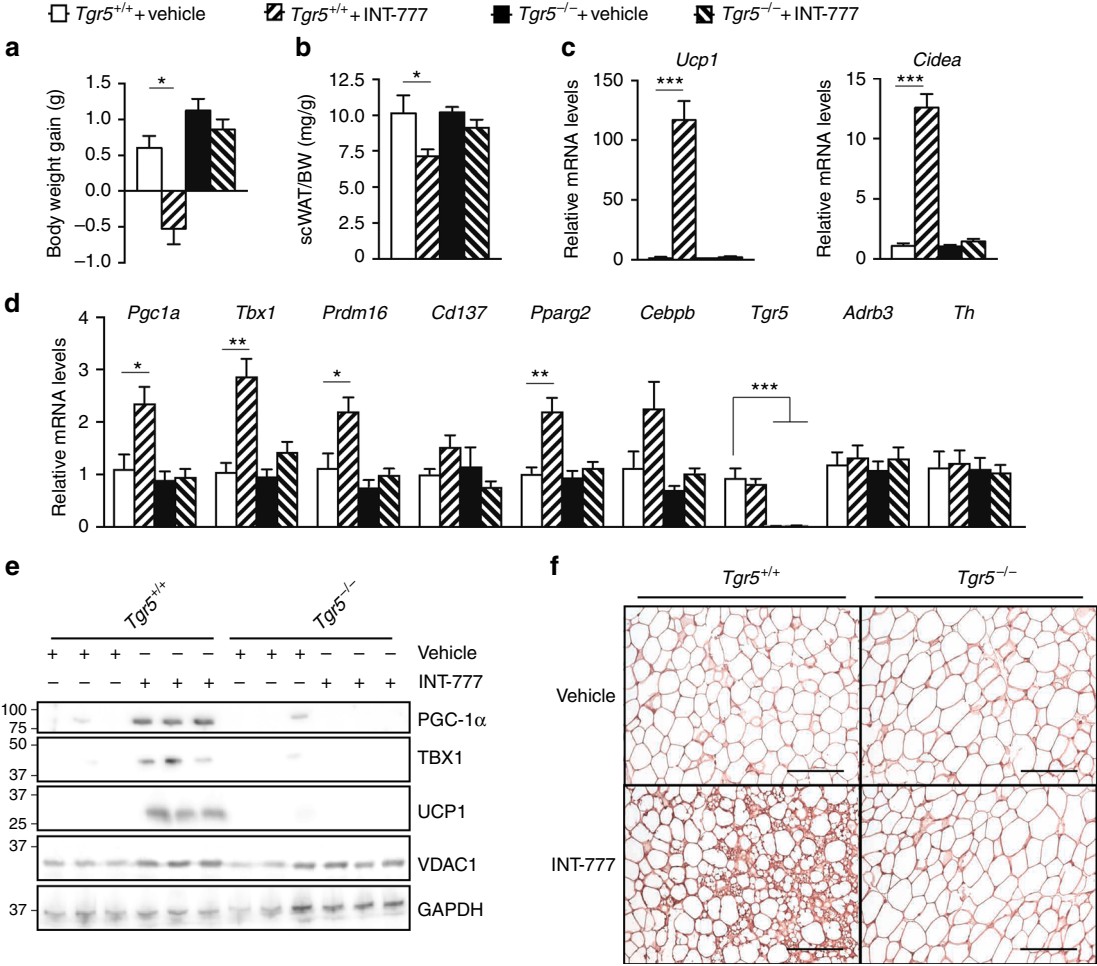

**Fig. 3** TGR5 activation induces scWAT beiging at thermoneutrality. **a** Body weight gain of TGR5 wild-type ($Tgr5^{+/+}$) and germline TGR5 knockout ($Tgr5^{-/-}$) mice housed at thermoneutrality (30 °C) and subjected to a daily administration of the selective TGR5 agonist INT-777 or vehicle for 7 days. $n = 10$ per group. **b** scWAT over body weight (BW) ratio of the mice described in **a**. **c**, **d** mRNA levels of beige remodelling markers *Ucp1* and *Cidea* (**c**), *Pgc1a*, *Tbx1*, *Prdm16*, and *Cd137*, *Pparg2* and *Cebpb*; *Tgr5* and adrenergic related genes (*Adrb3* and *Th*) (**d**) in the scWAT of mice described in **a**. **e** Representative ($n = 10$ per group) western blot of PGC-1α, the mitochondrial marker VDAC1, and beiging markers TBX1 and UCP1 from the scWAT of mice described in **a**. GAPDH was used as loading control. **f** Representative ($n = 5$ per group) UCP1 immunostaining of scWAT sections from mice described in **a**. Scale bars = 50 μm. Results represent mean ± SEM. *$P \leq 0.05$, **$P \leq 0.01$ and ***$P \leq 0.001$ vs. $Tgr5^{+/+}$ + Vehicle group by one-way ANOVA followed by Bonferroni post hoc test. Uncropped western blots are provided in Supplementary Fig. 11A–D

Supplementary Fig. 4B) and mtDNA/nDNA ratio (Fig. 4d). Finally, histomorphological analysis of the scWAT confirmed that INT-777 supplementation prevented adipocyte hypertrophy (Fig. 4e) supported by a reduction in adipocyte area (Fig. 4f) and induced the appearance of UCP1-positive brown-like adipocytes (Fig. 4g, h). These results demonstrate that activation of TGR5 promotes scWAT beiging also in the context of HF diet-induced obesity.

**TGR5 activation promotes beige remodelling in human cells.** In order to assess the relevance of our findings in human biology, we analysed the expression data from human bone marrow-derived mesenchymal stem cells induced for adipogenic differentiation[32]. *Tgr5* expression increased during the late stages of adipogenesis in conjunction with other adipocyte differentiation and beige remodelling markers (Fig. 5a), suggesting that TGR5 is involved in the differentiation process of human pre-adipocytes. Furthermore, exposure of INT-777 to human Simpson Golabi Behmel Syndrome (SGBS) pre-adipocyte cells[33–35] amplified the beige remodelling phenotype compared to vehicle (DMSO) (Fig. 5b). Together, these data indicate that TGR5 is expressed in

human adipocytes and that its activation leads to beige cell remodelling.

**TGR5 signalling induces mitochondrial biogenesis and function.** We then wished to elucidate the mechanism by which TGR5 elicits beiging of scWAT. For this purpose, adipose tissue-derived stromal vascular fraction (SVF) cells were isolated from subcutaneous fat pads of $Tgr5^{+/+}$ and $Tgr5^{-/-}$ mice and differentiated into adipocytes in the presence of INT-777 or DMSO for 7 days. Exposure to INT-777 induced several beige adipocyte markers in $Tgr5^{+/+}$ cells both at the transcript (Fig. 5c) and protein level (Fig. 5d, e). Differentiation into beige adipocytes by INT-777 was also confirmed in 3T3-L1 cells, in which activation of endogenous TGR5 (sh-Co) was sufficient to induce the expression of several beige cell markers including *Ucp1*, *Cidea*, *Pparg2* and *Cebpb*, whereas the increase of *Pgc1a* required TGR5 overexpression (mTGR5) (Supplementary Fig. 5A). Moreover, stimulation with the secondary BA lithocholic acid (LCA), a potent endogenous TGR5 agonist, also triggered an increase in *Pgc1a*, *Ucp1* and *Cidea* expression, that was blunted in cells in which TGR5 was silenced (sh-TGR5) (Supplementary Fig. 5B).

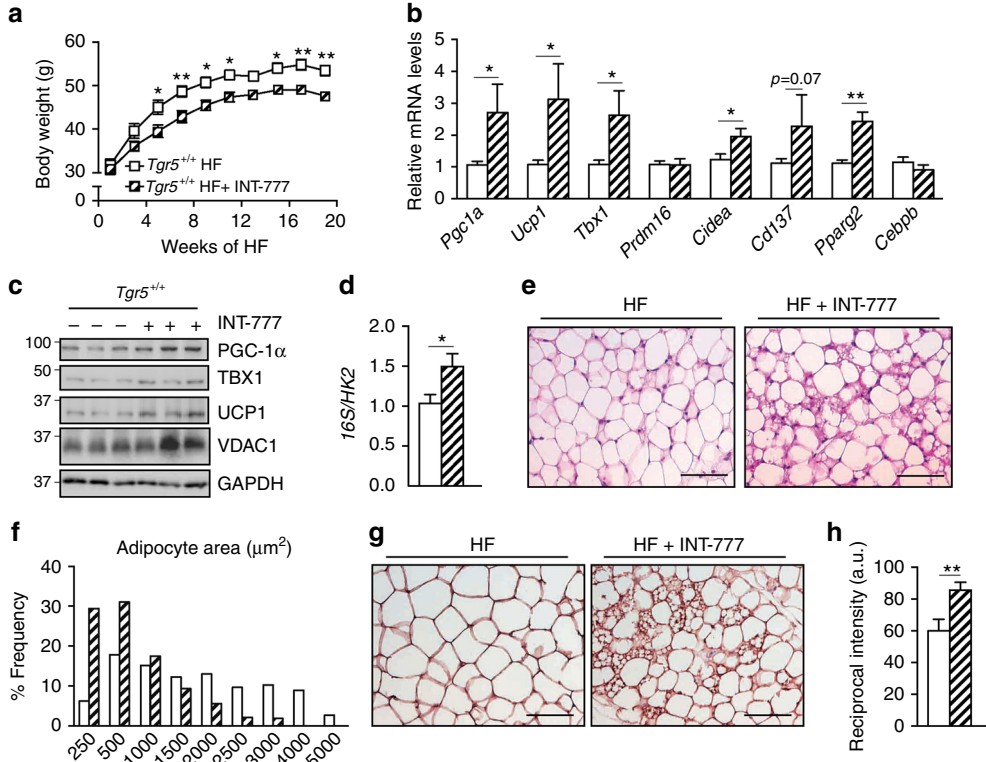

**Fig. 4** TGR5 activation induces scWAT beiging in mice fed a high-fat diet. **a** Body weight curves of TGR5 wild-type (*Tgr5*$^{+/+}$) fed a high-fat (HF) diet for 20 weeks in the presence or absence of the selective TGR5 agonist INT-777. $n = 10$ per group. **b** mRNA levels of beige remodelling markers *Pgc1a*, *Ucp1*, *Tbx1*, *Prdm16*, *Cidea*, *Cd137*, *Pparg2* and *Cebpb* in the scWAT of mice described in **a**. **c** Representative ($n = 10$ per group) western blot of PGC-1α, the mitochondrial marker VDAC1, and beiging markers TBX1 and UCP1 from the scWAT of mice described in **a**. GAPDH was used as loading control. **d** Quantification of mitochondrial (16S) vs. nuclear (HK2) DNA ratio from the scWAT of mice described in **a**. **e** Representative haematoxylin and eosin staining from the scWAT of mice described in **a**. **f** Adipocyte area quantification from images shown in **e**. **g** Representative ($n = 5$ per group) UCP1 immunostaining of scWAT sections from mice described in **a**. **h** Quantification of UCP1 immunostaining intensity depicted in **g**. Scale bars = 50 μm. Results represent mean ± SEM. *$P \leq 0.05$ and **$P \leq 0.01$ vs. *Tgr5*$^{+/+}$ HF group by two-way ANOVA followed by Bonferroni post hoc test (**a**) or Student's *t* test (**b**, **d**, **h**). Uncropped western blots are provided in Supplementary Fig. 12A and B

Both INT-777 and LCA enhanced UCP1 immunostaining (Supplementary Fig. 5C), as evidenced by the increase in relative fluorescence intensity (Supplementary Fig. 5D).

In line with the in vivo experiments, INT-777 treatment of primary adipocytes impacted both the expression of beige markers and mitochondrial abundance. Indeed, mtDNA/nDNA ratio (Fig. 5f), the mitochondrial markers VDAC1, translocase of outer mitochondrial membrane 40 (TOMM40) (Fig. 5d, e) and TOMM20 (Supplementary Fig. 5E, F) remained low in *Tgr5*$^{-/-}$ cells, but were significantly increased by TGR5 stimulation in *Tgr5*$^{+/+}$ adipocytes. These findings support the previous results that TGR5 is required for mitochondrial biogenesis, a process intrinsically linked to the beiging phenotype. To evaluate if the increase in mitochondrial content also impacts mitochondrial function, we performed respirometry assays. TGR5 activation increased spare respiratory capacity of *Tgr5*$^{+/+}$ INT-777 stimulated adipocytes after exposure to the mitochondrial uncoupler carbonyl cyanide 4-(trifluoromethoxy) phenylhydrazone (FCCP) (Fig. 5g). We then investigated if the improved mitochondrial function was accompanied by an increase in fatty acid oxidation and if this effect could be mediated by a modulation of substrate availability. Notably, activation of TGR5 signalling robustly induced lipolysis in *Tgr5*$^{+/+}$ but not in *Tgr5*$^{-/-}$ differentiated adipocytes, demonstrated by a marked increase in glycerol (Fig. 5h) and FFA (Fig. 5i) release. Moreover, the oxygen consumption rate (OCR) of *Tgr5*$^{+/+}$ adipocytes after INT-777 stimulation was significantly blunted after pre-incubation with the carnitine palmitoyltransferase-1 (CPT-1) inhibitor, etomoxir (Fig. 5j), which prevents the formation of acyl-carnitines, a necessary step for the transport of fatty acyl chains into the mitochondria. These results indicate that TGR5 activation improves mitochondrial function and promotes release of fatty acids, providing substrates for mitochondrial β-oxidation.

**TGR5 induces ERK/DRP1-dependent mitochondrial fission.** Mitochondrial remodelling is important for cellular bioenergetics[36] and could be a key process to establish the functional effects of beiging. To study the role of TGR5 in modulating mitochondrial morphology, we analysed the mitochondrial architecture and evaluated mitochondrial circularity from fluorescent micrographs of *Tgr5*$^{+/+}$ and *Tgr5*$^{-/-}$ preadipocytes and adipocytes stained with TOMM20 antibody after INT-777 exposure. Acute activation of TGR5 transformed the mitochondrial network composed of elongated mitochondria into fragmented mitochondria in both *Tgr5*$^{+/+}$ preadipocytes (Fig. 6a) and mature adipocytes (Fig. 6c). This was evidenced by a significant increase in total circularity (Fig. 6b, d), indicative of an increase in mitochondrial fission. To characterise the downstream signalling components involved in this process, we measured the phosphorylation of potential TGR5 targets in differentiated *Tgr5*$^{+/+}$ and *Tgr5*$^{-/-}$ adipocytes exposed for 30 min to INT-777. In line with the notion that TGR5 activation increases cAMP levels, CREB phosphorylation was induced in response to INT-

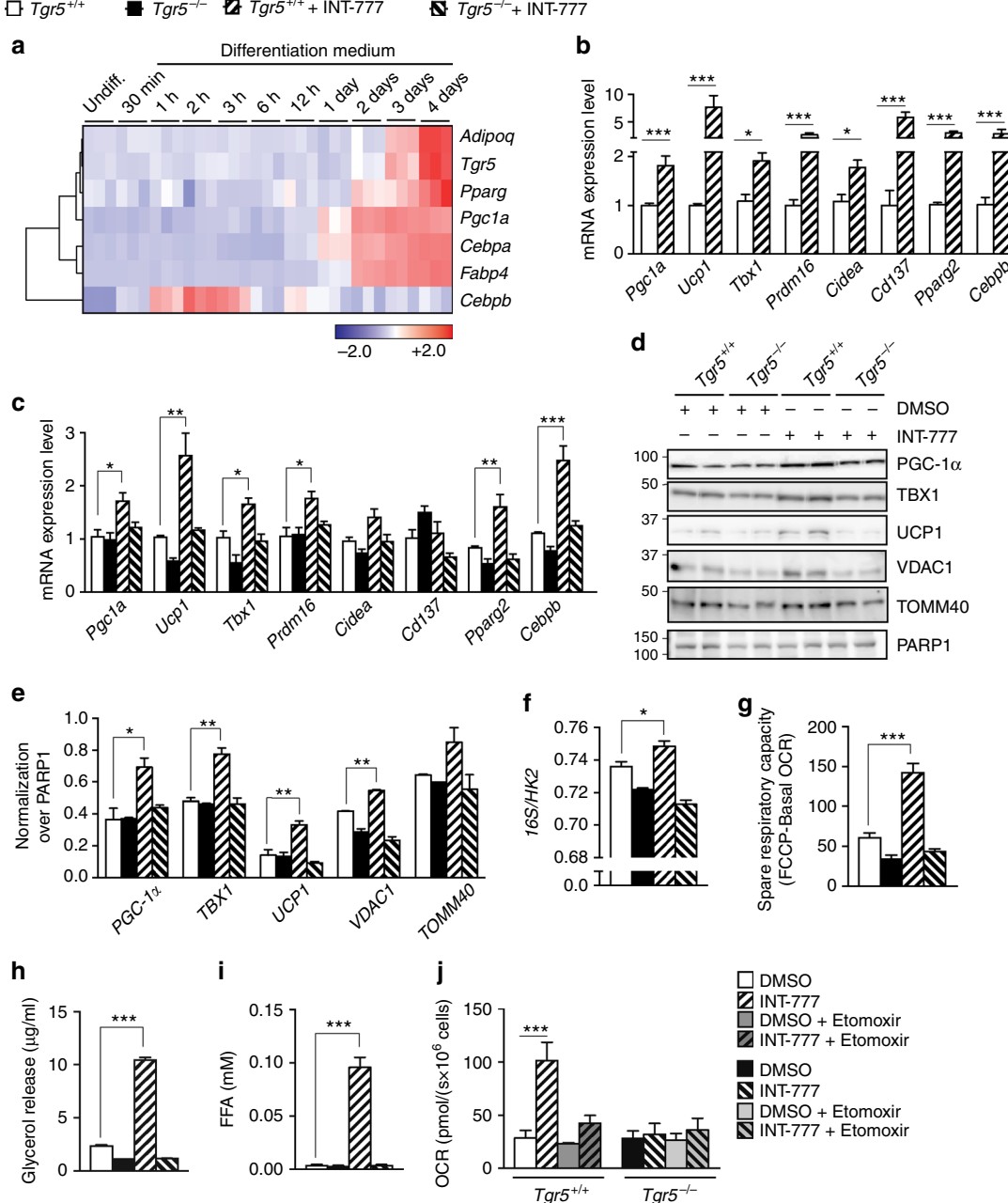

**Fig. 5** TGR5 activation promotes beige adipocyte differentiation in vitro. **a** Heat map showing the expression of *Tgr5* and adipocyte and beiging markers from human bone marrow-derived mesenchymal stem cells induced for adipogenic differentiation (GEO Accession number GSE80614)[32]. Colour key represents row *z*-score. **b** mRNA levels of beige remodelling markers *Pgc1a*, *Ucp1*, *Tbx1*, *Prdm16*, *Cidea*, *Cd137*, *Pparg2* and *Cebpb* assessed in the human pre-adipocyte cell line, Simpson Golabi Behmel Syndrome (SGBS). SGBS cells were differentiated in presence or absence of the TGR5 agonist INT-777. *n* = 6. **c** mRNA levels of genes described in **b** in adipocytes differentiated from the stromal vascular fraction (SVF) of TGR5 wild-type (*Tgr5*$^{+/+}$) and germline TGR5 knock-out (*Tgr5*$^{-/-}$) mice. SVF cells were differentiated for 7 days in presence or absence of the TGR5 agonist INT-777. *n* = 6. **d** Representative (*n* = 6 per group) western blot of PGC-1α, mitochondrial markers (VDAC1 and TOMM40) and beiging markers TBX1 and UCP1 from the cells described in **c**. PARP1 was used as loading control. **e** Quantitative densitometry of the western blots showed in **d**. **f** Quantification of mitochondrial (16S) vs. nuclear (HK2) DNA ratio from the cells described in **c**. **g** Spare respiratory capacity of the cells described in **c**, calculated as the difference between maximal (FCCP) and basal oxygen consumption rate (OCR). **h**, **i** Glycerol (**h**) and fatty acid (**i**) release from the cells described in **c** after 1 h stimulation with the TGR5 agonist INT-777 or vehicle (DMSO). **j** Basal oxygen consumption rate (OCR) of cells described in **c** after 3 h pre-incubation with etomoxir and stimulation with the TGR5 agonist INT-777 or vehicle (DMSO). Results represent mean ± SEM. *$P \leq 0.05$, **$P \leq 0.01$ and ***$P \leq 0.001$ vs. *Tgr5*$^{+/+}$ cells by one-way ANOVA followed by Bonferroni post hoc test (**c**, **e**–**j**) or Student's *t* test (**b**). Uncropped western blots are provided in Supplementary Fig. 13A–C and Supplementary Fig. 14A

777 in *Tgr5*$^{+/+}$ adipocytes (Fig. 6e; Supplementary Fig. 6A). However, the extracellular signal-regulated kinase (ERK) signalling pathway was also robustly activated (Fig. 6e; Supplementary Fig. 6A). One of the main proteins implicated in mitochondrial

fission is the GTPase dynamin-1-like protein (DRP1) that can be activated through phosphorylation by ERK at Serine 616 in mouse and human cells[37,38]. Conversely, phosphorylation of DRP1 at Serine 637 by PKA inactivates the protein leading to

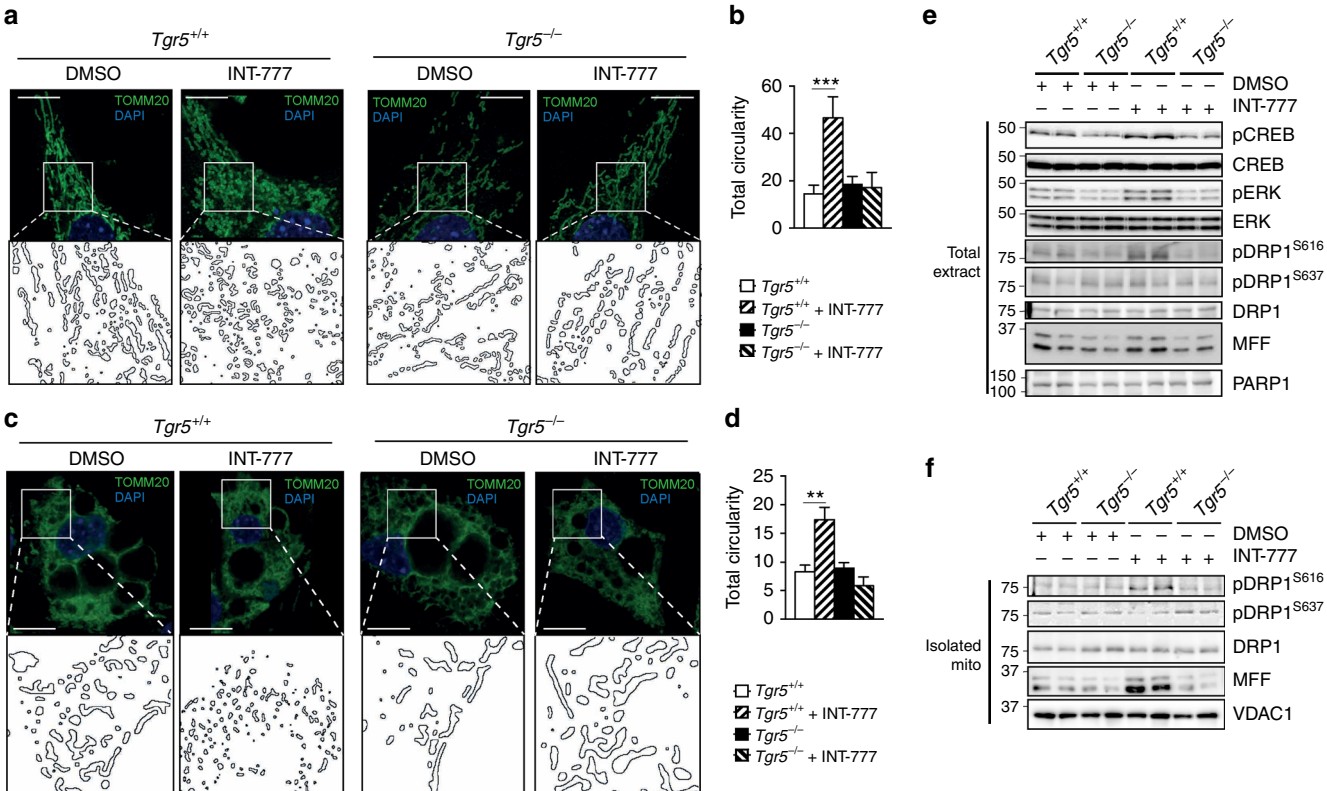

**Fig. 6** TGR5 activation induces mitochondrial fission in an ERK-DRP1-dependent manner. **a**, **c** Representative (*n* = 6 per group) images of TOMM20 immunofluorescence (in green) on preadipocytes (**a**) and differentiated adipocytes (**c**) derived from the stromal vascular fraction (SVF) of TGR5 wild-type (*Tgr5*+/+) and germline TGR5 knockout (*Tgr5*−/−) mice and stimulated with the TGR5 agonist INT-777 or vehicle (DMSO). Nuclei were stained with DAPI (in blue). Scale bars = 10 μm (**a**) 25 μm (**c**). Insets show higher magnification of the reconstructed mitochondrial network (ImageJ program). **b**, **d** Quantification of mitochondrial circularity from images as in **a** and **c** calculated with ImageJ. *n* = 6. **e**, **f** Representative (*n* = 6 per group) western blots of TGR5 downstream targets (phospho proteins), their relative controls (CREB, ERK and DRP1) and the mitochondrial protein MFF from the total extract (**e**) or isolated mitochondria (**f**) derived from differentiated adipocytes described in **c**. PARP1 (**e**) and VDAC1 (**f**) were used as loading controls. *n* = 6. Results represent mean ± SEM. **P ≤ 0.01 and ***P ≤ 0.001 vs. *Tgr5*+/+ cells by one-way ANOVA followed by Bonferroni post hoc test. Uncropped western blots are provided in Supplementary Fig. 13D and E and Supplementary Fig. 14B–E

mitochondrial fusion[39–41]. Of interest, TGR5 stimulation in *Tgr5*+/+ adipocytes increased DRP1 phosphorylation specifically at Serine 616, but not at Serine 637 (Fig. 6e; Supplementary Fig. 6A), suggesting that ERK signalling could be involved in the regulation of TGR5-mediated mitochondrial fission. Notably, we also observed an increase in the mitochondrial fission factor (MFF) protein in *Tgr5*+/+ adipocytes exposed to INT-777 (Fig. 6e; Supplementary Fig. 6A). MFF is a mitochondrial outer membrane protein that recruits DRP1 to the mitochondria to promote fragmentation[42] and changes in its expression can modulate mitochondrial remodelling[43,44]. We next assessed at what stage in the adipocyte differentiation process mitochondrial fission can be induced by TGR5 activation. For this purpose, we performed a kinetic assay in *Tgr5*+/+ and *Tgr5*−/− SVF cells differentiated into adipocytes for 1, 3, 5 and 7 days in the presence or absence of INT-777 or vehicle. Although CREB phosphorylation was increased by INT-777 stimulation during the whole differentiation process, ERK phosphorylation did not change until day 3 of differentiation, which coincided with an initial increase in mitochondrial abundance using VDAC1 immunoblotting as a marker (Supplementary Fig. 7A–H). DRP1 phosphorylation was significantly increased in adipocytes stimulated with INT-777 at days 5 and 7 of differentiation, as was the mitochondrial content marker VDAC1 (Supplementary Fig. 7E–H). These results suggest that the regulation of mitochondrial fission after activation of the TGR5-ERK signalling pathway goes in parallel with an induction in mitochondrial biogenesis, most likely orchestrated

by cAMP, which is an established inducer of genes involved in mitochondrial biogenesis[45,46]. We then isolated mitochondria from *Tgr5*+/+ and *Tgr5*−/− differentiated adipocytes and analysed DRP1 phosphorylation. Acute TGR5 stimulation induced DRP1 phosphorylation specifically at Serine 616, but not at Serine 637 (Fig. 6f; Supplementary Fig. 6B). Moreover, MFF was markedly enhanced in differentiated *Tgr5*+/+, but not *Tgr5*−/− adipocytes upon INT-777 exposure (Fig. 6f; Supplementary Fig. 6B). Taken together, these results show that acute activation of TGR5 signalling modulates the mitochondrial network by inducing mitochondrial fission through the activation of ERK-DRP1 signalling in primary adipocytes.

**ERK inhibition ablates TGR5-mediated mitochondrial fission.** To determine if the enhanced ERK signalling induced by TGR5 agonists is responsible for the mitochondrial fission phenotype, *Tgr5*+/+ and *Tgr5*−/− preadipocytes were differentiated in the absence or presence of the ERK inhibitor FR180204 and INT-777. Interestingly, PGC-1α and VDAC1 proteins (Fig. 7a; Supplementary Fig. 8A), as well as the mtDNA/nDNA ratio (Supplementary Fig. 8B), remained increased upon INT-777 stimulation, even when co-exposed with the ERK inhibitor. These results are consistent with the finding that CREB phosphorylation, a critical regulator of mitochondrial biogenesis, was not downregulated (Fig. 7a; Supplementary Fig. 8A). On the contrary, DRP1 phosphorylation at Serine 616 was lost upon ERK inhibition (Fig. 7a;

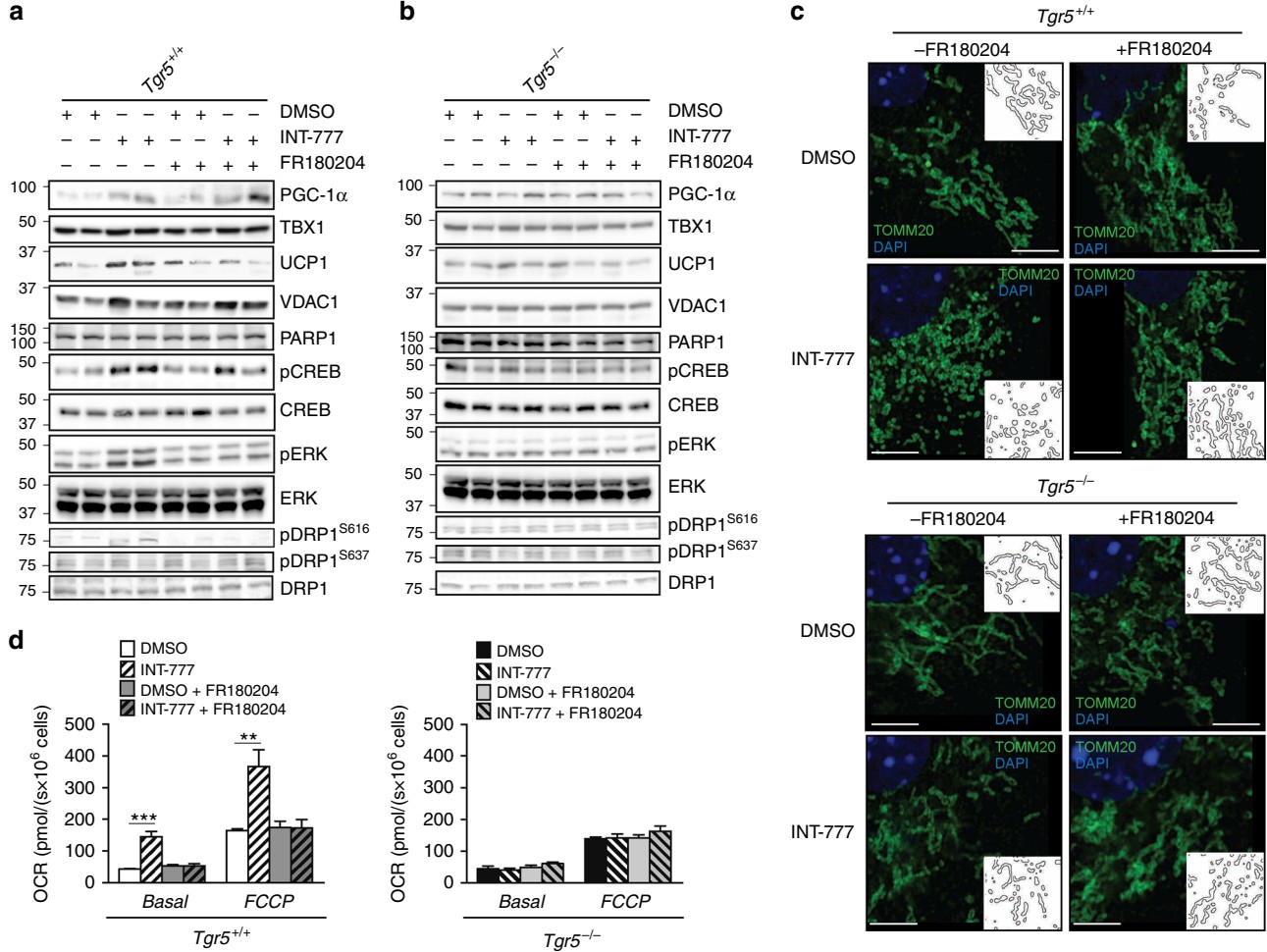

**Fig. 7** ERK activation is required for TGR5-mediated mitochondrial fission. **a**, **b** Representative ($n = 6$ per group) western blots of PGC-1α, the mitochondrial marker VDAC1, and beiging markers TBX1 and UCP1, and TGR5 downstream targets (phospho proteins) with their relative controls (CREB, ERK and DRP1) from differentiated adipocytes derived from the stromal vascular fraction (SVF) of TGR5 wild-type ($Tgr5^{+/+}$) (**a**) and germline TGR5 knockout ($Tgr5^{-/-}$) (**b**) mice. PARP1 was used as loading control. Cells were stimulated with the TGR5 agonist INT-777 or vehicle (DMSO) in the presence or absence of the selective ERK inhibitor FR180204. $n = 6$. **c** Representative ($n = 6$ per group) images of TOMM20 immunofluorescence (in green) on preadipocytes derived from the stromal vascular fraction (SVF) of TGR5 wild-type ($Tgr5^{+/+}$) and germline TGR5 knockout ($Tgr5^{-/-}$) mice. Cells were stimulated as described in **a** and **b**. Nuclei were stained with DAPI (in blue). Scale bars = 10 μm. Insets show a reconstruction of the mitochondrial network. $n = 6$. **d** Oxygen consumption rate (OCR) of the cells described in **a** and **b**. Cellular respiration was measured in basal condition (Basal) and at maximal respiration (FCCP). Results represent mean ± SEM. **$P \leq 0.01$ and ***$P \leq 0.001$ vs. $Tgr5^{+/+}$ cells by one-way ANOVA followed by Bonferroni post hoc test. Uncropped western blots are provided in Supplementary Fig. 15A–E and Supplementary Fig. 16A–E

Supplementary Fig. 8A), confirming that ERK is critical in mediating DRP1-dependent fission. In contrast, $Tgr5^{-/-}$ cells showed no induction of proteins involved in the beiging program and failed to induce ERK-DRP1 signalling after INT-777 (Fig. 7b; Supplementary Fig. 8A), indicating that DRP1 activation by ERK is fully dependent on TGR5.

To demonstrate that the changes observed in mitochondrial morphology were mediated through ERK, we analysed the mitochondrial network of $Tgr5^{+/+}$ and $Tgr5^{-/-}$ cells shortly treated with INT-777 in the absence or presence of FR180204. $Tgr5^{+/+}$ cells in which ERK signalling was blunted were unable to enhance mitochondrial fission compared to $Tgr5^{+/+}$ cells stimulated with INT-777 (Fig. 7c). This effect was dependent on TGR5 as no fragmentation could be observed in the $Tgr5^{-/-}$ cells (Fig. 7c). Since mitochondrial fragmentation is associated with maximal respiratory rates and increased proton conductance[47], we next assessed OCR in differentiated $Tgr5^{+/+}$ and $Tgr5^{-/-}$ adipocytes exposed to INT-777 alone or combined with FR180204. Acute induction of TGR5 signalling in $Tgr5^{+/+}$ cells

significantly increased OCR (Fig. 7d), both in basal and FCCP-uncoupled conditions, as well as the spare respiratory capacity (Supplementary Fig. 8C). However, this effect was lost in $Tgr5^{-/-}$ adipocytes or after selective ERK inhibition with FR180204 in $Tgr5^{+/+}$ adipocytes (Fig. 7d; Supplementary Fig. 8C), indicating that acute activation of TGR5 in primary adipocytes improves respiratory function by promoting mitochondrial fission in an ERK-dependent manner. Thus, ERK activation by TGR5 triggers the dynamics of mitochondria by mainly affecting the process of fission during beige adipocyte differentiation.

## Discussion

The consumption of high-caloric diets and the reduction in physical activity has resulted in a sharp increase in the prevalence of obesity and associated metabolic diseases. Increasing the activity of brown fat, beige fat or both hence holds great promise as a treatment to limit the progression of these chronic disorders[7,48]. Here we demonstrate that activation of the BA

responsive TGR5 signalling axis promotes the recruitment of thermogenic beige adipocytes within the scWAT, highlighting a novel mechanism by which BAs or TGR5 agonists may limit body weight gain. Mechanistically, our data support a model in which TGR5 activation triggers lipolysis, mitochondrial biogenesis and DRP1-mediated fission, three processes critically required to promote beiging and thermogenesis of white fat depots.

Beiging of scWAT has been shown to be orchestrated by various cell types residing within the white fat depot, including (pre-)adipocytes[49–51], sympathetic neuronal fibres[52,53] and immune cells[54–56]. We characterised a mouse strain with a loss-of-function of TGR5 in adipocytes and performed comprehensive studies in 3T3-L1 and primary beige cells. Both in vivo and in vitro models corroborate the importance and requirement of TGR5 in adipocytes to elicit beiging. Indeed, deletion of TGR5 in adipocytes significantly blunted the development of beige cells triggered via cold exposure. Accordingly, SVF cells lacking TGR5 were unable to differentiate into beige adipocytes, formed less mitochondria and displayed a lower respiration rate. Moreover, selective activation of TGR5 by INT-777 induced scWAT beige remodelling at thermoneutrality. This result goes in line with a recent study that showed that BAs can induce UCP1 expression at thermoneutrality in the BAT[57], indicating that TGR5-mediated beiging could be independent of cold-activated sympathetic activity. This would concur with our data that TGR5 agonists in combination with cues other than cold exposure, like for instance chronic HF diet feeding, also dramatically modulate beiging. Currently, it remains challenging to distinguish physiological effects of the BAT vs. the scWAT when studying a beiging factor expressed in both fat depots. Despite this limitation, the combination of the current in vivo and in vitro findings unequivocally establish BAs as agents that shift energy storing white adipocytes into metabolically active cells in which lipolysis and mitochondrial activity are enhanced. Further studies in scWAT-specific $Tgr5^{-/-}$ mouse models will be required to dissect the exact physiological contribution of these beige cells in terms of energy expenditure and body weight control.

Mitochondrial remodelling comprises changes in mitochondrial morphology and architecture[47], which are important for cell viability and senescence, and for mitochondrial health, bioenergetic function, and intracellular signalling[58,59]. One of the main findings emerging from this study is that TGR5 activation is critically required for driving the full remodelling process of beige fat differentiation. Our data showed that this occurs not only by coordinating the biogenesis and architecture of mitochondria, but also by enhancing their thermogenic capacity, and further consolidates the role of TGR5 as a key molecular target to modulate mitochondrial function.

The role of TGR5 in regulating mitochondrial function was previously studied in the BAT, where its activation increases deiodinase 2 expression to catalyse the bioactivation of thyroid hormone and to induce mitochondrial fat oxidation in brown adipocytes[19]. It was also reported that stimulation of the TGR5 signalling pathway activates mitochondrial oxidative phosphorylation and increases the ATP/ADP ratio in enteroendocrine L cells[26]. Despite these initial clues, this is the first time that TGR5 signalling is described to modulate mitochondrial morphology to increase cellular respiration. In the BAT, mitochondrial fission is used as a compensatory mechanism to activate thermogenesis under high nutrient supply by increasing fatty acid oxidation and promoting 'nutrient wasting' in the form of heat[36,47]. Our findings show that the activation of ERK/DRP1 signalling by TGR5 agonists induces mitochondrial fission as a mechanism to increase respiration in primary adipocytes. TGR5 signalling furthermore provoked a massive induction of lipolysis and FFA release in primary beige adipocytes. This

observation, most likely the consequence of cAMP/PKA induction, is fascinating, especially because FFA not only serve as substrates for β-oxidation, but also promote thermogenesis through UCP1 activation[60,61], and facilitate mitochondrial fission[36,47,62]. It is hence conceivable that lipolysis is an important process that contributes to the higher respiration rate and fragmented mitochondrial shape induced by FFA availability.

This study identified beiging of white adipose fat depots as a novel route by which BAs or TGR5 selective BA mimetics induce fat oxidation. As for all beiging pathways identified in mice, extrapolation of mouse studies to human biology remains challenging. A data set analysis performed in this study revealed that TGR5 is present in human adipocytes and its expression increases along the adipocyte differentiation process. Notably, SGBS cells, considered to be a representative in vitro model of human white pre-adipocytes[34], showed distinct features of beiging remodelling when differentiated in the presence of TGR5 activators, suggesting that our findings could be potentially translated to the clinic. Until now, the majority of human studies have focused on the modulation of circulating BAs to confer improvements in body weight control and glucose metabolism[63–66] but did not evaluate the target tissues affected by altered BA signalling. One earlier study reported the use of the BA chenodeoxycholic acid (CDCA) to promote mitochondrial uncoupling and energy expenditure via TGR5 in human BAT[28]; however, a more detailed analysis of the WAT beiging in vivo in humans is still lacking and the role of the other BAs that could activate TGR5 is unclear. Future studies are warranted to evaluate the use of TGR5 agonists in humans to assess the potency of these molecules in improving mitochondrial dynamics and respiration rate in white fat.

## Methods

**Study design**. The primary objective of this study was to investigate the role of TGR5 signalling as a novel pathway important for the induction of scWAT beiging. In vivo studies were performed in different mouse models of WAT beiging development, such as diet-induced obesity and prolonged cold exposure. We included cohorts of TGR5 wild-type ($Tgr5^{+/+}$) and TGR5 germline knockout ($Tgr5^{-/-}$) mice in addition to floxed TGR5 ($Tgr5^{Adipoq+/+}$) mice and tamoxifen-inducible adipose tissue-specific $Tgr5^{-/-}$ mouse strain ($Tgr5^{Adipoq-/-}$). Mice exposed to thermoneutrality (30 °C) were used as controls. Mice were randomised into the different groups according to the genotype. To explore the molecular mechanisms through which TGR5 can promote the development of beige adipocytes, we used an in vitro model of adipocyte precursors isolated from scWAT SVF of $Tgr5^{+/+}$ and $Tgr5^{-/-}$ mice, as well as the 3T3-L1-mouse fibroblast cell line and the human SGBS cell line.

The sample size was calculated on the basis of the known variability for each assay. A power analysis was performed to calculate the sample size for mouse experiments. All mouse experiments were performed once. Mice showing any sign of severity, predefined by the Veterinary Office of the Canton of Vaud, Switzerland (authorisation no. 2614 and 3077), were killed and excluded from the data analyses. These criteria were established before starting the experiments. Experiments in cell models were performed at least three times. In order to focus on TGR5-specific signalling and avoid confounding effects of FXR activation, the selective TGR5 agonist INT-777 was used for all experiments instead of natural BAs[67]. All experiments were performed in a non-blinded manner.

**Generation of the $Tgr5^{Adipoq-/-}$ mice**. The inducible adipose tissue-specific $Tgr5^{-/-}$ mice were generated by breeding mice carrying floxed alleles for the TGR5 gene ($Tgr5^{fl/fl}$ mice) with the tamoxifen-inducible adiponectin ($Adipoq$)CreER[T2] Cre mouse line (kind gift from Christian Wolfrum, ETH Zurich). For the validation, two groups of $Tgr5^{Adipoq+/+}$ and $Tgr5^{Adipoq-/-}$ 8-week-old male mice (10 mice per group) were treated with 1 mg tamoxifen for 5 consecutive days to ensure recombination in adipocytes. Successful spatial knockout of TGR5 mRNA expression in WAT was carefully assessed.

**Prolonged cold exposure**. Two groups of 10 mice (either $Tgr5^{+/+}$ and $Tgr5^{-/-}$ mice or $Tgr5^{Adipoq+/+}$ and $Tgr5^{Adipoq-/-}$) were exposed to prolonged cold test (7 days, 8 °C) at the Phenotyping Unit in EPFL. The mice were housed in groups of three mice per cage with access to food and water. Rectal measurements of body temperature were performed daily during the week of cold exposure. Mice were killed on the 7th day of cold exposure in normal feeding conditions (9:00 a.m.). The mice were anaesthetised by isoflurane inhalation, blood was collected by cardiac

puncture, and tissues were harvested, weighed, and flash-frozen in liquid nitrogen or fixed with 4% formalin to perform biochemical and histological analyses, respectively.

**Thermoneutrality controls**. Four groups of 10 mice (either $Tgr5^{+/+}$ and $Tgr5^{-/-}$ mice or $Tgr5^{Adipoq+/+}$ and $Tgr5^{Adipoq-/-}$) were housed in a 30 °C cabinet for 1 week and used as controls for the prolonged cold exposed mice. Mice were killed on the 7th day of 30 °C housing in normal feeding conditions (9:00 am) anaesthesia protocol and organ collection was performed as mentioned above for the prolonged cold exposure mice.

**Thermoneutrality and INT-777 administration**. Four groups of 10 mice ($Tgr5^{+/+}$ and $Tgr5^{-/-}$ mice) were housed at thermoneutral conditions (30 °C) for 2 weeks. The first week was used for adaptation and blunting of the sympathetic response. During the second week of thermoneutrality, mice were gavaged once per day with 60 mg/kg of INT-777 or vehicle (carboxyl methyl cellulose). Body weight was measured before and after treatment. Mice were killed after 7 days of INT-777 or vehicle delivery in normal feeding conditions, blood and organs were collected as previously mentioned for the prolonged cold exposure and thermoneutrality control mice.

**Diet-induced obesity model**. Eight-week-old $Tgr5^{+/+}$ (2 groups of 10 mice each) were fed with a HF diet (D12492; Research Diet) for 20 weeks, supplemented or not with the TGR5-specific agonist INT-777 (30 mg/kg). Body weight was monitored weekly. After 20 weeks of diet, mice were fasted overnight and were killed under isoflurane anaesthesia. Blood and metabolic tissues were collected as described above for the prolonged cold exposure and thermoneutrality mice models.

**Ethical approval**. All the mice experiments were authorised by the Veterinary Office of the Canton of Vaud, Switzerland under the license authorisations no. 2614 and 3077.

**Isolation and differentiation of adipocyte precursors**. Subcutaneous WAT (scWAT-inguinal) was dissected from 8 to 12 weeks old $Tgr5^{+/+}$ and $Tgr5^{-/-}$ mice. Lymph nodes were carefully removed and the tissue was subsequently digested with collagenase type II (1 mg/ml) for 45 min at 37 °C with shaking. Digestion was stopped by adding 15 ml of complete medium (Dulbecco's Modified Eagle's medium (DMEM) containing 10% foetal calf serum (FBS) and penicillin/streptomycin (PS—10 mg/ml)) followed by centrifugation at $700 \times g$ for 10 min. The pellet was resuspended, filtered through a 70 µm nylon mesh and centrifuged again. SVF cells were resuspended in red blood cell lysis buffer (154 mmol/l $NH_4Cl$, 10 mmol/l $KHCO_3$ and 0.1 mmol/l EDTA) for 10 min followed by centrifugation at $700 \times g$ for 10 min. Cells were resuspended, plated and cultured until confluence in DMEM, including glucose (4.5 g/l), 10% FBS and PS (10 mg/ml). Differentiation of precursor cells into adipocytes was induced for 3 days with DMEM, including glucose (4.5 g/l), 10% FBS, PS (10 mg/ml), insulin (6.063 µg/ml), dexamethasone (1 µM), and 3-isobutyl-1-methylxanthine (IBMX—0.5 mM). Cells were maintained for 4 consecutive days in DMEM, including glucose (4.5 g/l), 10% FBS, PS (10 mg/ml) and insulin (6.063 µg/ml). Cells were treated with the TGR5 agonist INT-777 (30 µM), the ERK inhibitor FR180204 (10 µM) or vehicle (dimethyl sulfoxide— DMSO) since the first day of differentiation. The medium was changed every 2 days.

For acute stimulation of TGR5 signalling pathway, low-glucose DMEM (1 g/l) FBS-free medium with 0.1% bovine serum albumin (BSA) was used for 4 h in pre-adipocytes and/or adipocytes to normalise cells to basal levels, followed by INT-777 or vehicle (DMSO) treatment for 30 min for protein collection or 1 h for TOMM20 immunocytochemistry to evaluate the mitochondrial network.

For lipolysis assessment, differentiated adipocytes were stimulated acutely (60 min) with either DMSO or INT-777 and incubated in 1 ml of medium (DMEM GIBCO 21063 Invitrogen) containing 0.1% BSA. After 1 h of incubation at 37 °C, the medium was collected and glycerol or FFA concentrations were measured using a free glycerol determination kit (Sigma-Aldrich) or a non-esterified fatty acid assay (Wako), respectively, accordingly to the manufacturer's instructions.

To evaluate the effect of TGR5 activation in fatty acid oxidation, differentiated $Tgr5^{+/+}$ and $Tgr5^{-/-}$ adipocytes were pretreated with the CPT-1 inhibitor etomoxir (50 µM), for 3 h in the presence and absence of INT-777 or vehicle (DMSO). Basal respirometry was assessed as specified below.

**Cell culture and transfection**. The 3T3-L1 mouse fibroblast cell line was obtained from the American Type Culture Collection (CL-173). 3T3-L1 fibroblasts were cultured in DMEM, including glucose (4.5 g/l), 10% FBS and PS (10 mg/ml). Overexpression of mouse TGR5 (mTGR5) or silencing with sh-TGR5 or control (sh-Co) were performed by transfecting the cells using Lipofectamine 2000 (Invitrogen) with 2.5 µg DNA per well in a six-well plate. Transfection of the cells was performed twice, before differentiation and at day 4 of differentiation. Differentiation of 3T3-L1 cells into adipocytes was induced for 3 days in DMEM, including glucose (4.5 g/l), 10% FBS, PS (10 mg/ml), insulin (6.063 µg/ml), dexamethasone (1 µM), and IBMX (0.5 mM). Cells were then maintained for 4

consecutive days in DMEM, including glucose (4.5 g/l), 10% FBS, PS (10 mg/ml) and insulin (6.063 µg/ml). Cells were treated with INT-777 (30 µM), litocholic acid (LCA 10 µM) or dimethyl sulfoxide (DMSO) as vehicle since the first day of differentiation.

SGBS human pre-adipocyte cell line from scWAT were a kind gift from Prof. Martin Wabitsch. SGBS cells were cultured until confluent in basal medium (DMEM/F12, Panthotenat/Biotin (1.7/3.3 mM), 10 mg/ml penicillin/streptomycin and 10% FBS), quick-differentiated for 3 days in Quick-diff medium (basal medium supplemented with 0.01 mg/ml transferrin, 20 nM insulin, 100 nM cortisol, 0.2 nM T3, 25 nM dexamethasone, 250 µM IBMX and 2 µM rosiglitazone), and then differentiated for 2 more weeks in 3FC medium (basal medium supplemented with 0.01 mg/ml transferrin, 20 nM insulin, 100 nM cortisol and 0.2 nM T3), following the protocol optimised in Prof. Wabitsch's laboratory. INT-777 or DMSO were added from the beginning of differentiation and medium was changed every 2 days.

Cells were tested for mycoplasma using MycoProbe (#CUL001B, R&D Systems), following the manufacturer's instructions.

**Quantitative real-time qPCR for mRNA and DNA quantification**. RNA was extracted from scWAT, epidymal WAT, BAT or in vitro differentiated adipocytes using TRIzol and was transcribed to complementary DNA using QuantiTect Reverse Transcription Kit (Qiagen) following manufacturer's instructions. Expression of selected genes was analysed using the LightCycler 480 System (Roche) and SYBR Green chemistry. All quantitative polymerase chain reaction (PCR) results were presented relative to the mean of acidic-ribosomal phosphoprotein PO gene ($36b4$) and beta-2-microglobulin gene ($B2m$) for mice samples and to mean of hypoxanthine phosphoribosyltransferase 1 ($Hprt1$) for human cells (DDCt method). The average of at least three technical repeats was used for each biological data point.

To measure mtDNA content (marker of mitochondrial number), genomic DNA was extracted from cultured differentiated adipocytes or from scWAT of $Tgr5^{+/+}$, $Tgr5^{-/-}$, $Tgr5^{Adipoq+/+}$ and $Tgr5^{Adipoq-/-}$ mice using the NucleoSpin Tissue kit (Macherey-Nagel, Germany) following manufacturer's instructions. Then, quantitative PCR was performed using mitochondrial DNA (16S) and nuclear DNA- (hexokinase 2 ($Hk2$)) specific primers. The primers sequences used for qPCR are available in Supplementary Table 1.

Heat map was generated using R and was derived from data deposited under GSE80614 Geo data sets repository[32].

**Western blotting**. Samples were lysed in lysis buffer (50 mM Tris (pH 7.4), 150 mM KCl, 1 mM EDTA, 1% NP-40, 5 mM NAM, 1 mM sodium butyrate, protease and phosphatase inhibitors). Proteins were separated by SDS–PAGE and transferred onto nitrocellulose or polyvinylidene difluoride membranes. Blocking and antibody incubations were performed in 5% BSA or non-fat dry milk. PGC-1α (ab54481 1:1000), UCP1 (ab10983 1:1000 for animal tissues, 1:500 for primary culture), TBX1 (ab109313 1:1000) and VDAC1 (ab14734 1:1000) antibodies were from Abcam; anti-CREB (9197 1:1000), p-CREB Ser133 (9191 1:1000), ERK (4695 1:1000), p-ERK (4376 1:1000), DRP1 (4E11B11 1:1000), p-DRP1 Ser616 (3455 1:1000) and p-DRP1 Ser637 (4867S 1:1000) antibodies were from Cell Signalling; anti-MFF antibody (17090-1-AP 1:1000) was from Proteintech group; anti-PARP1 (sc-7150 1:1000), GAPDH (sc-47724 1:2000) and TOMM40 (sc-11414 1:1000) antibodies were from Santa Cruz Biotechnology. The MitoProfile Total OXPHOS Rodent WB Antibody Cocktail (MS604-300 1:500) for mitochondrial subunits was purchased from MitoSciences. Antibody detection reactions were developed by enhanced chemiluminescence (Advansta) and imaged using the c300 imaging system (Azure Biosystems). Quantification was done using ImageJ software.

**Morphologic analysis**. scWAT, epidymal WAT and BAT samples were fixed in buffered formalin (4%) overnight and embedded in paraffin. About 5 µm thick serial sections were made from paraffin-embedded tissue and then stained with haematoxylin and eosin. A Leica DM500 light microscope (Leica Microsystems, Wetzlar Germany) was used for imaging. Quantification of adipocyte area was performed with Adiposoft automated software. Briefly, all the adipocytes were manually selected using ImageJ's multipoint selection and were separated by using the Watershed transform plugin. Adipocyte area and number were measured as described[68].

**Immunohistochemistry**. About 5 µm thick sections were made from paraffin-embedded scWAT, epiWAT or BAT. Sliced tissue was dewaxed, rehydrated and quenching was performed with 3% $H_2O_2$ in PBS 1×. Heat-induced epitope retrieval was performed in 10 mM citrate buffer pH 6 at 95 °C for 20 min. Blocking and antibody incubations were performed in 3% BSA. Blocking was for 1 h at room temperature (RT) and UCP1 primary antibody (U6382 from Sigma 1:500) was incubated overnight at 4 °C. Secondary antibody (anti-rabbit HRP-conjugated) was incubated for 1 h at RT. Enzymatic revelation was performed with 3,3′-diaminobenzidine tetrahydrochloride (HRP-DAB) followed by nuclear staining using Mayer's haematoxylin. A Leica DM500 light microscope (Leica Microsystems, Wetzlar Germany) was used for imaging. Quantification of chromogen intensity of immunohistochemistry was performed via reciprocal intensity method as using

ImageJ software. We split the image using the colour deconvolution tool and selecting the chromogen used for the staining (DAB). After measuring the intensity of the stained region of interest (ROI) in the DAB channel, we subtracted this value from 250, deriving a reciprocal intensity that is directly proportional to the amount of chromogen present. As previously reported[69], standard colour images acquired from bright field microscopy have a maximum intensity of value 250 (represented by white, unstained areas). Stained areas, as marked by DAB have lower intensity values, resulting in an inverse correlation between the amount of antigen and its numerical value.

**Immunocytochemistry.** Differentiated adipocytes were rinsed with 1× ice-cold PBS and fixed with 4% paraformaldehyde for 15 min at RT. After three washes, the cells were permeabilised with 0.1% Triton X-100 solution for 15 min at RT. Blocking and antibody incubation were performed in 3% BSA. Blocking was for 1 h at RT, and TOMM20 (sc-11415 1:200) or UCP1 (ab10983 1:500) primary antibodies were incubated overnight at 4 °C. Three washes were performed before incubation with secondary anti-rabbit Alexa Fluor488 (ab150077 1:250) antibody for 1 h at RT. Cells were washed, incubated with DAPI dye for 10 min and mounted. The imaging was performed with Zeiss LSM 700 confocal microscope (Carl Zeiss Microscopy, Germany) and analysis of the mitochondrial network and circularity was done using ImageJ software using macro designed by Ruben K. Dagda at the University of Pittsburgh (2010). Quantification of immunofluorescence was performed with ImageJ software. The ROI was delimited followed by splitting the image into the two-colour channels. We determine the average number of cells by measuring the integrated density value (IDV) for the blue channel (DAPI staining). Ten representative nuclei were selected to calculate the mean nucleus value and the IDV of the blue channel was divided by the mean nucleus value to obtain the average number of cells in each ROI. Finally, the IDV of the green channel was divided by the average number of cells represented as protein content per nucleus[70].

**Respirometry on differentiated adipocytes.** Differentiated adipocytes from $Tgr5^{+/+}$ and $Tgr5^{-/-}$ mice were used to measure OCR using high-resolution respirometry (Oxygraph-2k, OROBOROS Instruments). Cells were detached, counted, resuspended in DMEM (200,000 cells/ml) and loaded in the Oroboros chamber to measure basal respiration followed by titration of the protonophore agent FCCP (0.2 mM) to achieve maximum electron transfer flux.

**Statistics.** Differences between two groups were assessed using two-tailed $t$ tests. Differences between more than two groups were assessed using one-way analysis of variance (ANOVA). To compare the interaction between two factors, two-way ANOVA tests were performed. ANOVA, assessed by Bonferroni's post hoc test, was used when comparing more than two groups. GraphPad Prism 6 was used for all statistical analyses. All $P$-values < 0.05 were considered significant. $*P \leq 0.05$, $**P \leq 0.01$ and $***P \leq 0.001$.

**Data availability**. Data supporting the findings of this study are available within the paper and its supplementary information files. Data are also available from the corresponding author at request.

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

## Acknowledgements

We thank Dr. Christian Wolfrum for providing the (*Adipoq*)CreER$^{T2}$ mouse line, Prof. Martin Wabitsch for providing the SGBS cells, Prof. Roberto Pelliciari for kindly providing INT-777, and Norman Moullan, Andréane Fouassier, Sabrina Bichet, Thibaud Clerc, Roxane Pasquettaz, Aurelie Demierre and the Phenotyping Unit (UDP) for technical assistance. This work was funded by the Swiss National Science Foundation SNSF 31003A_125487 and the Ecole Polytechnique Fédérale de Lausanne (EPFL). L.A.V.-V. is supported by a postdoctoral fellowship from CONACYT (No. 263859) and the Foundation for Health and Education Dr. Salvador Zubirán A.C. M.Z. is supported by the KNOW consortium 'Healthy Animal—Safe Food' MS&HE No. 05-1/KNOW2/2015 and the Foundation for Polish Science (FNP). V.L. was supported by a PhD grant from the Portuguese Foundation for Science and Technology (SFRH/BD/52046/2012) through the Graduate Program in Basic and Applied Biology (GABBA) PhD program.

## Author contributions

L.A.V.-V., A.P. and K.S. conceived and designed the project. L.A.V.-V., A.P., V.L., M.Z., M.N. and T.W.H.P. performed experiments. L.A.V.-V. and A.P. analysed the data. L.A.V.-V. and K.S. wrote the manuscript with contributions from A.P.

## Additional information

**Competing interests:** The authors declare no competing financial interests.

