## [Peer Review File · Nature Communications]

Reviewers' comments:

Reviewer #1 (expert in bile acids and metabolism)(Remarks to the Author):

I read this manuscript few times with extreme interest. The authors did a great job. They underscored the putative role of TGR5 activation in the regulation of WAT energy expenditure. Indeed, they show that treating mice (or primary mouse cells) with the TGR5 specific INT777 ligand they can induce mitochondrial activation with energy uncoupling and the final beige remodeling of WAT. Few data are needed to further increase the quality of these novel and intriguing observations.

1. This reviewer is surprised that TGR5null mice on HF diet do not gain more weight than WT mice on HF diet. If the authors would plot the BW on figs 1a and 1b without INT777 maybe there is no difference at all. How does this loss of function finding reconcile with the new idea of TGR5 guiding the WAT beige remodeling? would this be only a putative pharmacological approach? also, on a different angle, absence of TGR5 in the BAT would not be important at all? The authors are invited to comment and to bring putative explanation also in respect of the literature. Furthermore, the question then becomes spontaneous. What is the physiological role of TGR5 in wat? First, what is the amount of TGR5 in wat? in the supplementary figures with TGR5 expression levels in different tissues, the authors are also invited to show the cycle times and eventually in situ hybridization in adipocytes.

2. The translational relevance is important. How would TGR5 expression levels change in wat of obese versus lean subjects? first of all, would TGR5 be there in obese WAT? If it is a target, it should be there. This is an important proof of concept for the present study. Would it be possible to reproduce the beige remodeling in human adipocytes? finally, is TGR5 expressed during the process of adipogenesis? and if so, would the process be modulated by INT777? IN which step of adipogenesis from fibroblasts to adipocytes TGR5 is getting expressed? Could the phenotype be related also to an anti-adipogenic effect?

3. According to the authors, the mechanism for the observed effect of INT777 is WAT beige remodeling via mitochondrial activity. Is fatty acid oxidation involved? or it is only uncoupling? would ATP increase in these cells and get coupled with oxygen consumption? the authors clearly show the increase in respiratory complexes content, nevertheless Oxygraphy is what they need to show. How is lipogenesis also changed? Obviously the best experiment would be the treatment with INT777 in presence of FAO inhibitors and in absence of mitochondrial DNA. Maybe these are very difficult to do, I do not know of availability mitoDNA depleted WAT cells, but certainly few functional experiments should be coupled with the gene expression modulation. The authors have the expertise and the superb capacity to design and perform them.

Reviewer #2 (expert in BAT and metabolism)(Remarks to the Author):

The manuscript entitled "TGR5 signalling promotes mitochondrial fission and beige remodelling of white adipose tissue" by Velazquez-Villegas et al. shows that chronic stimulation of the bile acid-responsive membrane receptor TGR5 promote beiging of the subcutaneous white adipose tissue (scWAT). Chronic administration of TGR5-selective bile acid mimetics induced browning as well as increased in mitochondrial content in scWAT of diet-induced obese mice. TGR5 also promoted cold-induced beiging in scWAT. This phenotype was recapitulated in vitro in differentiated adipocytes, where TGR5 activation promoted mitochondrial fission through the ERK/DRP1 pathway as well as improvement of mitochondrial respiration.

The novelty of the paper consists in the identification of TGR5 as a targetable scWAT browning factor, through the increase of adipocytes mitochondrial contents and the improvement of their mitochondrial respiration.

The manuscript is well written making it easy to read. However even though the assessment and analysis of how TGR5 activation promotes scWAT beiging through the regulation of mitochondrial remodelling and biogenesis is interesting, the manuscript at the moment lacks essential information regarding the in vivo metabolic relevance of the phenotyping of the TGR5 wt and knockout mice with respect to the challenges posed to the mice i.e HFD +/- INT-777 treatment, cold exposure, etc. The authors should provide data with respect of body weight, oxygen consumption etc...for these challenges. Experiments in this manuscript also lack clear controls as none of these had been made at thermoneutrality (30°C), or in chow diet conditions. Without this information the real contribution of scWAT browning mediated by TGR5 activation with respect to the thermogenic phenotype of the mice is impossible to evaluate, and as consequence also the relevance of the authors findings with respect to treatment of obesity and related metabolic complications.

Major comments

Results:

Fig 1. It would be informative if the authors provide a tissue distribution analysis of the mRNA expression of TGR5 in the different adipose tissue depots, especially BAT, scWAT and epiWAT side by side in a basal and a BAs or BAs mimetics stimulated condition. In the paper that the authors cite from their group i.e Watanabe et al, Nature, 2006 (ref.19 in the manuscript), there is a tissue distribution including epiWAT and BAT, showing that the expression of TGR5 at basal level is higher in BAT than in epiWAT. Because the authors look at the browning in the scWAT of the mice, it would be important to have the information also for the scWAT.

Fig.1a-b. The authors should include the body weight of the TGR5 wt and knockout mice on HFD either treated or not with INT-777 on the same graph, to see if the difference of weight between ko group and the wt are significant. Having the data in two different graphs makes difficult to pick up any significant difference between the two genotypes, rendering questionable the relevance of the browning effect that the authors see in their model with respect to the treatment of obesity and weight loss.

Fig1g. Authors claims that INT-777 supplementation prevented adipocyte hypertrophy. They should provide a quantification of the adipocytes size among different conditions to demonstrate this.

Fig.1h. The staining of UCP1 seems not very specific. Everything seems to have the same orange colour. Can the authors provide a better picture showing a clearer staining of UCP1 or use another staining method to get the same information? Moreover it would be nice to have also the quantification of the expression of UCP1 in the scWAT slides for the different experimental groups.

FigS1c and e. Differences between gene expression of PGC1a, PRDM16 and Tbx1 in epi WAT is observed, but not in proteins. Can the authors comment on that. Moreover, Tbx1 seems oversaturated. Can the authors provide a quantification of their western blot?

Fig.2. After one week of cold exposure, it does not seem that there is any difference of weight loss between ko and wt mice, so here the relevance of the beiging with respect to weight loss and obesity is questionable. Did the authors try a longer cold exposure? Or put the mice on HFD and then in the cold? Stressing a bit more the system could result in the enhancement of the differences between ko and wt mice.

Fig.2 and S1. The authors should show the same data for TGR5 wt and ko mice at thermoneutrality, which is the control for the cold exposed ones.

The authors mention in the text that they generated an inducible WAT-specific TGR5 knockout mouse to test the impact of TGR5 absence in adipocyte on the induction of beigeing using tan AdipQCreERT2 transgenic mouse. However adiponectin is not specific for WAT, also BAT expresses adiponectin, as is shown in Fig. S2. A better Cre model to use could have been the Prx1-Cre mouse that is supposed to be specific for scWAT. Concerning the use of the AdipoQ-CreERT2 mouse, the authors should be careful in drawing their conclusions and interpretation.

Fig2c. The authors claims that Tgr5^{+/+} mice but not Tgr5^{-/-} mice develops beigeing in scWAt, but there is no increase of Cidea, neither CD137. Can the authors comment on that? Moreover, there is an increase in PPAR γ expression. Could Tgr5 have an effect on adipose differentiation and not only on beigeing?

Fig2f. The decrease of CII and CIV observed in western blotting experiment is not obvious. Authors should provide a quantification to convince the reviewer.

Fig 3. The authors should present data about the body weight of the tissue-specific TGR5 knock-out mice vs wt. Again, without this information the relevance of the browning of the scWAT with respect of obese phenotype and weight loss is questionable. Moreover, at which age of the mice did the authors induce the Cre recombinase to get the deletion of the gene? Also, tamoxifen is known to have an impact on body weight, did the authors check if the administration of the tamoxifen to their model had any impact on their results?

Fig.S4c. The authors should provide a quantification of the UCP1 expression assessed by IF. They claim that LCA increases UCP1 expression vs DMSO control, but in these images it is not that evident.

Fig.4b. with respect to protein expression of TOMM40 (mitochondrial marker) in the TGR5 wt adipocytes treated with INT-777, should not it be increased? Why authors changed their control gene? Is there variation in GAPDH? PABP1 seems to be regulated in different conditions. Can the authors provide another control?

Fig.4e. In this panel is difficult to know to what we are looking at and for what reason we are looking at it. The authors should include in the panel that we are looking at TOMM20 expression if we have understood properly. Moreover it is difficult to see any difference between the experimental groups. A proper quantification of the intensity of the labelling between the different experimental groups is necessary to draw any conclusion.

Fig4f. There is no units on OCR calculation graph. Can the authors give this information? How do the authors normalised their data? Is it normalised to proteins? DNA?

FigS4. Can authors provide protein quantification of UCP1 in different condition to validate data in immunofluorescence?

Fig.S6c. The authors claim that after 3 days of differentiation the TGR5 wt adipocytes treated with INT-777 present increased pERK expression. However from the immunoblot it is difficult to conclude that. Moreover the TGR5 knockout adipocytes (treated or not with INT-777) at this stage show increased expression of total ERK vs controls. Can the authors comment on that? Moreover, total DRP1 expression is increased in TGR5 knockout adipocytes not treated with INT-777, but not pDRP1. Can the authors comment on that too?

Fig.S6. What about pCREB and CREB protein expression levels at the different stages of differentiation for the different experimental groups? Generally speaking the different proteins and phosphorylation site of the same proteins of the ERK-DRP1 signalling pathway that the authors analyse do not increase or decrease at the same stage of the differentiation. Have the authors an

explanation for that?

Minor comments

Results:

Fig1.c The authors measured the mRNA expression of pparg in the scWAT of the mice. They should provide the same information for pparg2, which is the isoform playing a key role in adipogenesis. Same for Fig.S1c-d, Fig.2c, FigS3b, FigS4a-b.

Fig6. Line 256 in the text fig 6d is cited but does not exist.

We thank the reviewers for their positive feedback and constructive comments. We have addressed all major concerns and performed a series of additional *in vitro* and *in vivo* experiments to complete the study according to the reviewers' suggestions. These data have been included and discussed in the revised manuscript (new parts highlighted in red).

The outcome consolidates our initial observations and unequivocally establishes the TGR5 signaling axis as a powerful circuitry to guide beige remodeling of the scWAT at thermoneutrality and in response to various environmental cues, such as cold exposure and high fat diet feeding. This work also identified mitochondrial fission as a critical process by which bile acids or TGR5 activators potentiate mitochondrial respiration. Within the context of beige fat remodeling, this mechanism is novel and is expected to contribute to the general understanding of the molecular mechanisms involved in beiging.

Answer to reviewers' comments:

Reviewer #1

1a. This reviewer is surprised that TGR5null mice on HF diet do not gain more weight than WT mice on HF diet. If the authors would plot the BW on figs 1a and 1b without INT777 maybe there is no difference at all. How does this loss of function finding reconcile with the new idea of TGR5 guiding the WAT beige remodeling?

We thank this reviewer for making this comment. There is indeed no significant difference in body weight gain between untreated *Tgr5*^{+/+} and *Tgr5*^{-/-} mice. This lack of effect can be due to several reasons, but does not necessarily reflect lack of physiological relevance of our observed phenotype, i.e. beige remodeling. In the past we have observed differences in body weight gain in mice fed a high fat-high sucrose (HF-HS) diet. In these experimental conditions, *Tgr5*^{-/-} mice gained more weight than their wild-type littermates (Rebuttal Figure 1). Unfortunately we did not collect the scWAT in this study to assess the beiging capacity. We do not know the exact cause for this discrepancy with the HF diet experiment in the current study, but we postulate that changes in the composition of the various high calorie diets may trigger the generation of agonists or provide metabolites that indirectly amplify the activity of TGR5. We are currently testing this hypothesis. As these studies will largely exceed the time allowed for the revision and to avoid confusion on this unresolved question, we decided to remove this part and hope to publish these data once we obtain more insight. However, we kept the pharmacological data of the HF diet study, which demonstrate the importance of TGR5 activators in triggering the beiging process (New Fig. 4) (Lines 168-178). In light of our new data demonstrating beige cell remodeling at thermoneutrality after only one week of agonist treatment (New Fig. 3) (Lines 145-159),

these data reinforce the notion that activation of the TGR5 signaling axis is essential for the beige cell remodeling.

Rebuttal Figure 1. Body weight gain in *Tgr5*^{+/+} and *Tgr5*^{-/-} mice after high fat diets feeding. Body weight of TGR5 wild-type (*Tgr5*^{+/+}) and germline TGR5 knock-out (*Tgr5*^{-/-}) mice fed a high fat (HF; D12492 Research Diets) diet (A) or a high fat high sucrose diet (HF-HS; TD08811 Harlan Laboratories) (B) for 12 weeks. *n* = 10 per group. Results represent mean ± SEM. * *P* ≤ 0.05 and ** *P* ≤ 0.01 vs. *Tgr5*^{+/+} group by two-way ANOVA followed by Bonferroni post-test.

1b. Would this be only a putative pharmacological approach?

As already pointed out above, we have included a new experiment under thermoneutral conditions (New Fig. 3) (Lines 145-159). These new data provide unequivocal evidence that activation of TGR5 induces beige remodeling in the scWAT even in the absence of adrenergic stimulation, indicating a functional and metabolic role of TGR5 in white adipose fat depots. From this experiment, it is clear that pharmacological activation of TGR5 triggers profound metabolic effects. However, based on our hypothesis that differences in the diet may alter TGR5 activity (see reply to question 1a), it is well conceivable that physiological variations in plasma bile acids may impact the process of browning. This is in line with previously published findings showing that cold exposure—a well-known inducer of the beiging process—also induces changes in BA composition, including secondary BAs, which are potent activators of TGR5 (Zietak, M. et.al. *Cell Metab*, 2016; Worthmann, A. et.al. *Nat Med*, 2017).

1c. Also, on a different angle, absence of TGR5 in the BAT would not be important at all? The authors are invited to comment and to bring putative explanation also in respect of the literature.

The exact contribution of the brown fat vs the beige fat is a very interesting question, which at this moment is difficult to address. We postulate that both tissues contribute to the anti-obesity effects initially observed, yet with the currently available models it is impossible to

quantify the exact contribution for each of these types of fat. Based on our current findings, we can only conclude that TGR5 in adipose depots contributes to an improved metabolic profile. It will be interesting to address this question once the appropriate tools will become available (for more details, see also reply to Reviewer 2, question Fig. 2).

1d. Furthermore, the question then becomes spontaneous. What is the physiological role of TGR5 in wat? First, what is the amount of TGR5 in wat? in the supplementary figures with TGR5 expression levels in different tissues, the authors are also invited to show the cycle times and eventually in situ hybridization in adipocytes.

TGR5 is a G protein-coupled receptor (GPCR), and like many receptors of this family, it is lowly expressed in most tissues (CT value between 30-32), except for the gallbladder (CT value of 26). When comparing TGR5 expression between the brown and white adipose tissues (Rebuttal Figure 2), we observed that TGR5 levels are lower in the epididymal WAT (epiWAT), however in the scWAT its expression is comparable to that found in BAT. In addition, administration of INT-777 does not have an effect on TGR5 expression, indicating that it is not modulation of its expression level, but rather its activation that is responsible to drive the metabolic effects in most of the tissues.

Rebuttal Figure 2. *Tgr5* expression in adipose tissue fat pads. *Tgr5* mRNA levels in brown adipose tissue (BAT), epididymal (epiWAT) and subcutaneous (scWAT) white adipose tissue of TGR5 wild-type (*Tgr5*^{+/+}) and germline TGR5 knock-out (*Tgr5*^{-/-}) mice fed a high fat (HF) diet for 20 weeks, in presence or absence of the selective TGR5 agonist INT-777. *n* = 10 per group. Results represent mean ± SEM. Different letters represent significance by two-way ANOVA followed by Bonferroni post-test. a>b>c.

Additionally, in this work, we demonstrate that one of the metabolic roles of TGR5 in the WAT is the modulation of energy substrates utilization by this tissue, i.e. promoting the mobilization and optimization of fatty acid oxidation by the mitochondria (See reply of question 3).

2. The translational relevance is important. How would TGR5 expression levels change in WAT of obese versus lean subjects? first of all, would TGR5 be there in obese WAT? If it is a target, it should be there. This is an important proof of concept for the present study. Would it be possible to reproduce the beige remodeling in human adipocytes? finally, is TGR5 expressed during the process of adipogenesis? and if so, would the process be modulated by INT777? IN which step of adipogenesis from fibroblasts to adipocytes TGR5 is getting expressed? Could the phenotype be related also to an anti-adipogenic effect?

We fully agree with this reviewer that translational relevance is important. Unfortunately, we did not get access to human biopsies from obese and lean subjects, but we now include data that support the presence of TGR5 in human adipocytes (New Fig. 5a). More specifically, we show that TGR5 transcript levels change along the differentiation process and become detected in human adipocytes starting from day 3 of adipogenesis (New Fig. 5a), paralleling the increase in the expression of other well-known adipocyte markers (Lines 182-186). In addition, adipocyte differentiation of the human Simpson Golabi Behmel Syndrome (SGBS) pre-adipocyte cell line in the presence of INT-777 amplifies the beige remodeling phenotype compared to vehicle (DMSO) (New Fig. 5b) (Lines 186-190) indicating that our findings could also be extrapolated to human biology.

It is unlikely that TGR5 is associated with an anti-adipogenic effect. In fact, the induction of Tgr5 expression is concomitant with other adipogenic markers (New Fig. 5a) (Lines 184-186). Our data rather indicate that activation of TGR5 induces a plethora of metabolic, transcriptional and structural processes characteristic of beige cell remodeling. A typical example is the enhanced lipolysis observed after TGR5 activation in mouse primary adipocytes. This promotes the release of glycerol and free fatty acids and increases their oxidation in the mitochondria leading to a more functional phenotype (New Fig. 5h-j) (Lines 218-227).

Finally, analysis from a microarray profiling database from scWAT adipocytes of obese and lean subjects (Lee, YH. *Diabetologia*, 2005) showed that TGR5 is also expressed in human white adipose tissue and increased in obese subjects compared to lean subjects (Rebuttal Figure 3). However, we opted not to include these data in the new version of the manuscript for two reasons: first, only very few beige markers could be detected in the array; second, we cannot exclude that the increase in TGR5 expression stems from enhanced recruitment of macrophages into the adipose tissue, which is a known phenomenon in the setting of obesity (Perino A. et.al. *J Clin Invest*, 2014). A dedicated study designed to address these shortcomings should be conducted to answer these questions.

Rebuttal Figure 3. Tgr5 expression in human white adipose tissue from lean and obese subjects. Analysis from a microarray database (Lee, YH. *Diabetologia*, 2005) showing *Tgr5* expression in subcutaneous white adipose tissue from obese and lean subjects. Results represent mean \pm SEM. *** $P \leq 0.001$ vs. lean subjects by Student t-test.

3. According to the authors, the mechanism for the observed effect of INT777 is WAT beige remodeling via mitochondrial activity. Is fatty acid oxidation involved? or it is only uncoupling? would ATP increase in these cells and get coupled with oxygen consumption? the authors clearly show the increase in respiratory complexes content, nevertheless Oxygraphy is what they need to show. How is lipogenesis also changed? Obviously, the best experiment would be the treatment with INT777 in presence of FAO inhibitors and in absence of mitochondrial DNA. Maybe these are very difficult to do, I do not know of availability mitoDNA depleted WAT cells, but certainly few functional experiments should be coupled with the gene expression modulation. The authors have the expertise and the superb capacity to design and perform them.

These are all insightful comments that are relevant to establish the physiological role of TGR5 in scWAT. To address these questions, we have performed a series of new experiments addressing the role of TGR5 in controlling fatty acid homeostasis. We now prove that induction of the TGR5 signaling pathway in mature adipocytes activates lipolysis, evidenced by an increase in glycerol (New Fig. 5h) and free fatty acid (New Fig. 5i) release after INT-777 stimulation (Lines 220-222). Moreover, activation of TGR5 increases fatty acid oxidation and this effect is blunted after pre-incubation of differentiated adipocytes with the CPT-1 inhibitor etomoxir (New Fig. 5j), indicating that the increase in mitochondrial respiration is mainly due to fatty acid oxidation (Lines 222-227). We opted to perform oxygraphy experiments upon TGR5 activation (New Fig. 5g and j) and in the presence or absence of FAO inhibitors (New Fig. 5j) to demonstrate that TGR5 stimulation in differentiated primary adipocytes is required to boost a fatty acid-dependent mitochondrial respiration. As mentioned by this reviewer, we agree that the use of mitoDNA depleted cells would be very interesting but challenging and probably not essential to characterize the importance of TGR5 in modulating the beiging process.

Reviewer #2:

The manuscript entitled “TGR5 signalling promotes mitochondrial fission and beige remodelling of white adipose tissue” by Velazquez-Villegas et al. shows that chronic stimulation of the bile acid-responsive membrane receptor TGR5 promote beiging of the subcutaneous white adipose tissue (scWAT). Chronic administration of TGR5-selective bile acid mimetics induced browning as well as increased in mitochondrial content in scWAT of diet-induced obese mice. TGR5 also promoted cold-induced beiging in scWAT. This phenotype was recapitulated in vitro in differentiated adipocytes, where TGR5 activation promoted mitochondrial fission through the ERK/DRP1 pathway as well as improvement of mitochondrial respiration.

The novelty of the paper consists in the identification of TGR5 as a targetable scWAT browning factor, through the increase of adipocytes mitochondrial contents and the improvement of their mitochondrial respiration.

The manuscript is well written making it easy to read. However even though the assessment and analysis of how TGR5 activation promotes scWAT beiging through the regulation of mitochondrial remodeling and biogenesis is interesting, the manuscript at the moment lacks essential information regarding the in vivo metabolic relevance of the phenotyping of the TGR5 wt and knockout mice with respect to the challenges posed to the mice i.e HFD +/- INT-777 treatment, cold exposure, etc. The authors should provide data with respect of body weight, oxygen consumption etc...for these challenges. Experiments in this manuscript also lack clear controls as none of these had been made at thermoneutrality (30°C), or in chow diet conditions. Without this information, the real contribution of scWAT browning mediated by TGR5 activation with respect to the thermogenic phenotype of the mice is impossible to evaluate, and as consequence also the relevance of the authors findings with respect to treatment of obesity and related metabolic complications.

We thank this reviewer for his/her helpful suggestions. We have performed a series of additional experiments to address the major concerns. More specifically, we have conducted additional *in vivo* studies at thermoneutrality and compared to cold-exposed TGR5 genetically engineered mouse models (GEMMs) (Data included in New Figs. 1 and 2). In addition, we have evaluated the beiging capacity of TGR5 activators at thermoneutrality. We demonstrate that 1 week of TGR5 activation at thermoneutrality is sufficient to induce scWAT beiging, and to promote body weight and scWAT loss (New Fig. 3). These additional studies consolidate our initial observations and unequivocally establish the TGR5 signaling axis as a novel circuitry to orchestrate beige fat remodeling

not only in response to cold exposure, but also at thermoneutrality in the absence of adrenergic stimulus. These data also further substantiate that TGR5 agonists, such as bile acids or bile acid mimicking agents, are absolutely required to drive the beiging process.

Based on this evidence, the question from this reviewer as to whether our findings are therapeutically relevant for the treatment of obesity can be distilled to the question “does beiging reduce body weight gain?”. Several studies in mice have been published that support the idea that beiging induction confers protection to obesity development (Kopecky J. et.al *J Clin Invest*, 1995; Wu J. et.al *Genes & Dev*, 2013; Kim, S.H. et.al. *Diabetes Metab J.*; Harms M, *Nat Med*, 2013). In addition, loss of beiging capacity gives insight into the pathogenesis of age-associated metabolic dysfunction (Rogers, N. et.al. *Aging Cell*, 2012; Mueller E. *Front Endocrinol*, 2016;). Some beiging agents have been found to exert a local effect on the WAT and are associated with increased energy expenditure and body weight loss including the fibroblast growth factor-21 (FGF21) (Fisher FM.et.al. *Genes & Dev*, 2012), cardiac natriuretic peptides (Lu X. et.al. *Endocrinology*, 2012), the myokine irisin (Bostrom, J. *Nature*, 2012) and the bone morphogenetic proteins BMP7 (Boon MR. et.al. *PLoS ONE*, 2013), BMP8B (Villarroya B. et.al. *Cell*, 2012) and BMP4 (Hoffman JM. et.al. *Cell Rep*, 2017). An important question that still remains unanswered in our study relates to the efficacy of the beige fat vs brown fat activity to burn excess of body fat. In this context, dissecting the exact contribution of the beiging phenotype from the other actions of TGR5 on body weight loss by analyzing scWAT-specific *Tgr5*^{-/-} mice will undoubtedly provide important insight. Unfortunately, this would require 2 years of work and as far as we know, there are currently no CRE drivers available that induce recombination only in the WAT (for more details see point-by-point reply below for Fig 2). Taken into account our findings and the observations from existing literature, we believe it is highly likely that the beiging phenotype contributes to the metabolic benefits of TGR5 activators. Yet, we will have to wait for the availability of scWAT-specific TGR5 GEMMs to obtain conclusive evidence.

Finally, as was suggested by this reviewer, we also compared beige signature markers in the scWAT of chow diet (CD) fed animals. Similar to what we observed in thermoneutral mice, chow fed mice did not express detectable levels of beiging markers and there is no significant difference between *Tgr5*^{+/+} and *Tgr5*^{-/-} mice, indicating once more that agonist stimulation is essential to trigger the induction of the thermogenic program in a TGR5 dependent manner. Unfortunately, we have not performed long-term studies with INT-777 supplementation under CD feeding as we did for the HF diet. Since these studies will largely exceed the time allowed for the revision we were not able to include these data. However, we expect to have similar results than those observed at thermoneutrality and under HF diet feeding after selective TGR5 activation. Given the substantial number of additional data in the new version of this manuscript, we have only retained the essential information that corroborates this finding (New Figs. 3 and 4).

Major comments

Results:

Fig 1. It would be informative if the authors provide a tissue distribution analysis of the mRNA expression of TGR5 in the different adipose tissue depots, especially BAT, scWAT and epiWAT side by side in a basal and a BAs or BAs mimetics stimulated condition. In the paper that the authors cite from their group i.e Watanabe et al, Nature, 2006 (ref.19 in the manuscript), there is a tissue distribution including epiWAT and BAT, showing that the expression of TGR5 at basal level is higher in BAT than in epiWAT. Because the authors look at the browning in the scWAT of the mice, it would be important to have the information also for the scWAT.

We analyzed *Tgr5* expression in brown (BAT) and white (WAT) adipose tissues. As previously reported (Watanabe et.al. Nature, 2006), TGR5 expression is lower in the epididymal WAT, however in the scWAT its expression is comparable to the one found in BAT. Administration of INT-777 does not have an effect on TGR5 expression, indicating that it is not a modulation in the expression of the receptor, but rather a change in its activation that is responsible for driving the metabolic effects in the adipose tissue (this information can be found in the reply to Reviewer 1, question 1d and Rebuttal Figure 2).

Fig.1a-b. The authors should include the body weight of the TGR5 wt and knockout mice on HFD either treated or not with INT-777 on the same graph, to see if the difference of weight between ko group and the wt are significant. Having the data in two different graphs makes difficult to pick up any significant difference between the two genotypes, rendering questionable the relevance of the browning effect that the authors see in their model with respect to the treatment of obesity and weight loss.

We understand the request of the reviewers to combine both panels. We did not present the body weight of *Tgr5*^{+/+} and *Tgr5*^{-/-} mice under HF diet in the same graph because it was difficult to interpret due to the fact that in this study no difference could be detected between the two genotypes without agonist treatment. However, we observed a significant difference in body weight between the *Tgr5*^{+/+} mice supplemented or not with the bile acid mimetic INT-777 that was not observed in *Tgr5*^{-/-} mice, indicating that the effect is relevant and is TGR5-specific. Consistent with the mechanism of action of GPCRs, this result points out that activation of TGR5 is essentially required to mediate its metabolic effects. The absence of a significant effect between the two untreated genotypes indirectly suggests that the activity of endogenous ligands of TGR5 is not strong enough to induce an effect on body weight. Alternatively, this particular HF diet, in contrast to other high calorie diets may provide factors that interfere with normal activation of bile acids (for

more details, see also reply to Reviewer 1, question Fig. 1a and Rebuttal Figure 1). Finally, it should be noted that in *Tgr5*^{-/-} mice, TGR5 is ablated in virtually every tissue, and this can have counteractive effects that may ultimately “neutralize” the effect on body weight. Further analysis is warranted to solve this conundrum. In view of the substantial number of additional data in the new version of this manuscript, we have decided to take out these confusing data and to retain only the most relevant information that HF diet feeding triggers beiging in a TGR5 dependent manner (New Fig. 4) (Lines 168-178).

Fig1g. Authors claims that INT-777 supplementation prevented adipocyte hypertrophy. They should provide a quantification of the adipocytes size among different conditions to demonstrate this.

We now provide a quantification of adipocyte size and show it in New Fig. 4f (Line 176).

Fig.1h. The staining of UCP1 seems not very specific. Everything seems to have the same orange color. Can the authors provide a better picture showing a clearer staining of UCP1 or use another staining method to get the same information? Moreover, it would be nice to have also the quantification of the expression of UCP1 in the scWAT slides for the different experimental groups.

To address this point, we repeated the UCP1 immunostaining using a different antibody (U6382 from Sigma) (Line 538) (New Fig. 4g). Additionally, we performed the quantification using the reciprocal intensity method (New Fig. 4h) as reported previously (Nguyen, D. 2013) (Lines 542-544).

FigS1c and e. Differences between gene expression of PGC1a, PRDM16 and Tbx1 in epi WAT is observed, but not in proteins. Can the authors comment on that. Moreover, Tbx1 seems oversaturated. Can the authors provide a quantification of their western blot?

We have re-analyzed the results from cold-exposed epiWAT and confirm the existing discrepancy between transcript and protein levels for a subset of beiging markers (New Supplementary Figure 1B). Overall, *Pgc1a*, *Tbx1* and *Prdm16* mRNA levels in epiWAT of wild-type mice were substantially lower (Cp values of 30-35) than those detected in scWAT (Cp values of 22-26) and did not result in changes at the protein level (New Supplementary Figure 1D and E) (Lines 108-114). The lack of effect can be explained by the fact that it is the scWAT and not the epiWAT in mice that preferentially undergoes induction of beiging (Guerra, C. *J Clin Invest*, 1998). A different regulation of mRNA and

protein cannot be excluded, however, the very low expression of these beige markers and the similar expression level at the protein levels underscore our initial findings. As requested by this reviewer, we now also included a less exposed Western blot for TBX1 and corresponding quantification (New Supplementary Fig. 1D and E).

Fig.2. After one week of cold exposure, it does not seem that there is any difference of weight loss between ko and wt mice, so here the relevance of the beigeing with respect to weight loss and obesity is questionable. Did the authors try a longer cold exposure? Or put the mice on HFD and then in the cold? Stressing a bit more the system could result in the enhancement of the differences between ko and wt mice.

Body weight loss between both genotypes did not reach significance, but there is a clear trend that *Tgr5*^{-/-} mice lose less weight after one week of cold exposure (New Fig. 1a) (Lines 86-89). We conjectured that this could be linked to the fact that the mice were not treated with TGR5 agonists. Indeed, if we treat *Tgr5*^{+/+} mice during one week with a TGR5 selective bile acid derivative, beigeing of the scWAT (New Fig 3c-f) is accompanied with a robust reduction in body weight gain and fat mass (New Fig 3a-b). These experiments were conducted at thermoneutrality, confirming the relevance of TGR5 activation in the control of body weight (New Fig. 3a).

Fig.2 and S1. The authors should show the same data for TGR5 wt and ko mice at thermoneutrality, which is the control for the cold exposed ones.

We now include in the manuscript data obtained from mice housed at thermoneutrality (30°C) for both *Tgr5*^{+/+} and *Tgr5*^{-/-} (New Fig. 1) (Lines 85-90 and 96-98) and *Tgr5*^{Adipoq^{+/+}} and *Tgr5*^{Adipoq^{-/-}} mice (New Fig. 2) (Lines 126-134).

The authors mention in the text that they generated an inducible WAT-specific TGR5 knockout mouse to test the impact of TGR5 absence in adipocyte on the induction of beigeing using tan AdipoQCreERT2 transgenic mouse. However, adiponectin is not specific for WAT, also BAT expresses adiponectin, as is shown in Fig. S2. A better Cre model to use could have been the Prx1-Cre mouse that is supposed to be specific for scWAT. Concerning the use of the AdipoQ-CreERT2 mouse, the authors should be careful in drawing their conclusions and interpretation.

It is true that with the animal models used in this study we are not able to unequivocally prove that the observed phenotype is specifically due to loss/activation of TGR5 in the scWAT, especially since we know that TGR5 is widely expressed and present in metabolic tissues such as the BAT. However, we have indications that the Prx1-Cre mice model is also not specific for scWAT. Prx1 is expressed in mesenchymal stem cells that contribute

to bone, muscle and both white and brown adipose tissue homeostasis (Calo, E. et.al. *Nature*, 2010). This Cre line is also widely used as a genetic mouse model for bone studies (Davey, R. et.al. *J Bone Miner Res*, 2004; Joshep, C. et.al. *Cell Stem Cell*, 2013; Elefteriou, F. et. al. *Bone*, 2011) and we know that TGR5 is also expressed in the bone (unpublished data). In addition, there is a strict interplay between bone and WAT and there is increasing evidence that bone acts as a novel metabolic organ.

We also agree with the reviewer that all the mouse models have limitations and thus, we need to be careful to avoid misinterpretations and erroneous conclusions. For this reason, we supported our findings with *in vitro* and *ex vivo* experiments, in which we demonstrated that activation of TGR5 in scWAT adipocyte precursors stimulates beige remodeling and enhances mitochondrial function. Although these data will not address the relative importance of the brown vs beige phenotype, they clearly underscore the importance of beiging as an additional mechanism that could underlie the anti-obesity effects mediated by bile acids (Watanabe M. et.al. *Nature*, 2006). Further studies will be needed to tease out the exact contribution of each of these fat depots once specific CRE drivers become available.

Fig2c. The authors claim that Tgr5^{+/+} mice but not Tgr5^{-/-} mice develops beiging in scWAt, but there is no increase of Cidea, neither CD137. Can the authors comment on that? Moreover, there is an increase in PPAR γ expression. Could Tgr5 have an effect on adipose differentiation and not only on beiging?

We thank the reviewer for this question. We were also puzzled by these unexpected data and have repeated the experiment with newly designed oligonucleotides. With these new primer sets, we could observe a significant difference in *Cidea* and *Cd137* transcript levels between *Tgr5*^{+/+} and *Tgr5*^{-/-} mice (New Fig. 1d and e), suggesting a problem with the previous set of primers. We also measured *Pparg2* expression in our settings. *Pparg2* levels were significantly increased by TGR5 activation in both *in vivo* and *in vitro* models. We included these new data in the manuscript and added the potential biological implications of these findings in the context of adipocyte differentiation (Lines 184-186).

Fig2f. The decrease of CII and CIV observed in western blotting experiment is not obvious. Authors should provide a quantification to convince the reviewer.

The quantification of the mitochondrial complexes can be found in New Supplementary Fig. 1A (Lines 102-103).

Fig 3. The authors should present data about the body weight of the tissue-specific TGR5 knock-out mice vs wt. Again, without this information the relevance of the browning of the scWAT with respect of obese phenotype and weight loss is

questionable. Moreover, at which age of the mice did the authors induce the Cre recombinase to get the deletion of the gene? Also, tamoxifen is known to have an impact on body weight, did the authors check if the administration of the tamoxifen to their model had any impact on their results?

Tamoxifen is a selective estrogen receptor modulator that is known to induce weight loss as a common side effect (Lampert C. *Physiol Behav*, 2013; Nisses, MJ. *Clin Breast Cancer*, 2011; Hesselbarth, N. et.al. *Biochem Biophys Res Commun*, 2015). As a consequence, measuring body weight loss is not a reliable readout for thermogenic activity in this model, even if tamoxifen is administered to $Tgr5^{Adipoq+/+}$ and $Tgr5^{Adipoq-/-}$ mice for the same period of time. To circumvent this confounding factor, we have measured body temperature as a more direct and functional readout for thermogenic activity. Interestingly, we observed that $Tgr5^{Adipoq-/-}$ mice could not cope with prolonged cold exposure and were unable to maintain their body temperature (New Fig. 2a) (Lines 128-131). Analysis of fat depots from these mice showed a robust impairment in scWAT beiging (New Fig. 2b-i). Although we cannot discern the exact contribution of the beige cells to this phenotype (TGR5 in brown fat is also reduced by $\pm 40\%$ in $Tgr5^{Adipoq-/-}$ mice (Suppl Fig. 2a), the effect on the BAT is less clear. While some of the markers (including PGC1 α) tend to be less expressed in the BAT of $Tgr5^{Adipoq-/-}$ vs $Tgr5^{Adipoq+/+}$ mice, others (e.g. UCP1) remain unchanged (New Supplementary Fig. 3C). Based on our new findings that point to a role of TGR5 in coordinating substrate availability for fatty acid oxidation (new Fig. 5h-j), it is possible that this may also be the case in BAT in which the increase in fatty acid levels could serve to activate UCP-1 mediated uncoupling (Rial E. et. al. *Eur J Biochem*, 1983; Shabalina I.G. et.al. *Biochim Biophys Acta*, 2008; Divakaruni A.S. et. al. *JBC*, 2012). Future studies in more appropriate models will provide more insight into this unresolved question.

Fig.S4c. The authors should provide a quantification of the UCP1 expression assessed by IF. They claim that LCA increases UCP1 expression vs DMSO control, but in these images it is not that evident.

We quantified UCP1 immunofluorescence in 3T3-L1 cells (New Supplementary Fig. 5D) (Line 206-207) and we observe a significant difference between INT-777 and LCA treatment compared to the DMSO control.

Fig.4b. with respect to protein expression of TOMM40 (mitochondrial marker) in the TGR5 wt adipocytes treated with INT-777, should not it be increased? Why authors changed their control gene? Is there variation in GAPDH? PABP1 seems to be regulated in different conditions. Can the authors provide another control?

We have used two mitochondrial proteins, TOMM40 and VDAC1 to assess the mitochondrial content after exposure to a TGR5 agonist. While VDAC1 levels are significantly increased, TOMM40 levels are not, but they clearly follow a similar trend (New Fig. 5d and e). In line with these data, mitochondrial DNA versus nuclear DNA ratio is significantly induced (New Fig. 5f). Together, these different approaches reinforce the notion that TGR5 activation increases the number of mitochondria in primary adipocytes. As for GAPDH, we did not use this protein as a loading control for the cells since we observed that it was changing between the different days of adipocyte differentiation. We hence opted for PARP1, which turned out to be a much better loading control for our *in vitro* experiments. In addition, we also have total CREB as control (New Fig. 6e, New Fig. 7a and b, and New Supplementary Figure 7).

Fig.4e. In this panel is difficult to know to what we are looking at and for what reason we are looking at it. The authors should include in the panel that we are looking at TOMM20 expression if we have understood properly. Moreover, it is difficult to see any difference between the experimental groups. A proper quantification of the intensity of the labelling between the different experimental groups is necessary to draw any conclusion.

We added to the figures the corresponding legend indicating TOMM20 immunofluorescence (Supplementary Fig. 5E, New Fig. 6a,c, and New Fig. 7c). We also performed the quantification of the fluorescence intensity normalized to cell content by DAPI staining and it is presented in New Supplementary Fig. 5F (Line 211).

Fig4f. There is no units on OCR calculation graph. Can the authors give this information? How do the authors normalised their data? Is it normalised to proteins? DNA?

The graph doesn't have units because it is a subtraction between maximal (FCCP) and basal respiration (New Fig. 5g). OCR was measured as indicated in lines 560-564 and the unit is $\text{pmol}/\text{s} \cdot 10^6$ cells (New Fig. 5j).

FigS4. Can authors provide protein quantification of UCP1 in different condition to validate data in immunofluorescence?

We performed the quantification of the fluorescence intensity normalized to cell content by DAPI staining in 3T3-L1 cells (New Supplementary Fig. 5D) (Lines 206-207).

Fig.S6c. The authors claim that after 3 days of differentiation the TGR5 wt adipocytes

treated with INT-777 present increased pERK expression. However from the immunoblot it is difficult to conclude that. Moreover the TGR5 knockout adipocytes (treated or not with INT-777) at this stage show increased expression of total ERK vs controls. Can the authors comment on that? Moreover, total DRP1 expression is increased in TGR5 knockout adipocytes not treated with INT-777, but not pDRP1. Can the authors comment on that too?

The quantification of the immunoblot indicates that pERK is significantly increased in *Tgr5*^{+/+} adipocytes treated with INT-777 after 3 days of differentiation (New Supplementary Fig. 7C and D). Total ERK and total DRP1 are probably increased in the *Tgr5*^{-/-} adipocytes as a compensation mechanism due to the fact that TGR5 is not expressed and is not activated, and ERK and DRP1 phosphorylation is reduced.

Fig.S6. What about pCREB and CREB protein expression levels at the different stages of differentiation for the different experimental groups? Generally speaking the different proteins and phosphorylation site of the same proteins of the ERK-DRP1 signalling pathway that the authors analyse do not increase or decrease at the same stage of the differentiation. Have the authors an explanation for that?

As suggested by this reviewer, we performed new Western blot analysis and checked CREB expression and phosphorylation during adipocyte differentiation. pCREB is significantly increased in *Tgr5*^{+/+} precursors after TGR5 stimulation since day 1 of differentiation, and is maintained until day 7, while total CREB amounts were not modified by INT-777 treatment (New Supplementary Fig. 7A-G) (Lines 258-259). As mentioned above, our data indicate that TGR5 is required for the adipocyte differentiation program. The variation of protein expression/phosphorylation in this kinetic experiment could be due to compensatory effects for the absence or stimulation of TGR5. The study of the precise role of TGR5 in mediating adipocyte differentiation is out of the scope of this manuscript and future studies will be required to unveil the precise molecular mechanisms beyond these TGR5-mediated effects.

Minor comments

Results:

Fig1.c The authors measured the mRNA expression of pparg in the scWAT of the

mice. They should provide the same information for pparg2, which is the isoform playing a key role in adipogenesis. Same for Fig.S1c-d, Fig.2c, FigS3b, FigS4a-b.

We measured *Pparg2* gene expression in all the *in vivo* and *in vitro* experiments: scWAT, epiWAT and BAT of cold (New Figs. 1 and 2, and New Supplementary Figs. 1 and 3) HF diet-fed (New Fig. 4), and thermoneutrality (New Fig. 3) exposed *Tgr5* GEMM mice and cells (primary (New Fig. 5) and 3T3-L1 (New Supplementary Fig. 5)).

Fig6. Line 256 in the text fig 6d is cited but does not exist.

We thank the reviewer for pointing this out. We have now added the label in New Fig. 7d corresponding to the OCR measurement in primary adipocytes in the presence or absence of INT-777 and the ERK inhibitor FR180204 (Line 297).

REVIEWERS' COMMENTS:

Reviewer #1 (Remarks to the Author):

This new version of the manuscript is stronger and the authors did a great job in the revision process.

Reviewer #2 (Remarks to the Author):

Comments to the manuscript NCOMMS-17-08665A

We appreciate the effort of the authors in answering the concerns we raised. However, the main question remains open: which is the contribution/relevance of TGR5 dependent scWAT browning and mitochondrial biogenesis on the overall thermogenic phenotype of the mice. The authors in a previous paper and a review suggest already the importance of mechanisms involving TGR5 for BAT activation and its contribution to modulate energy expenditure (Watanabe et al, Nature, 2006; Houten et al, Embo J, 2006), so what is the contribution of TGR5 activation to BAT thermogenesis vs scWAT browning with respect to the mouse metabolic phenotype, when cold exposed or fed an HFD is important. Moreover, because the authors claim that TGR5 could represent a targetable candidate to prevent obesity and related metabolic complication. All in all, the data presented about the mechanism by which TGR5 activation induce mitochondrial biogenesis and browning is interesting but do not answer the main question.

Major comments

Results

Fig.1a and 3a. The authors present the body weight gain of the mice, instead of the growth curve. Is it because the body weight of the mice were not similar between the experimental groups at the start of the experiment?

Fig 1a-3a. As mentioned in their answer to the reviewer comments, the authors designed a new set of oligonucleotides for Cidea and CD137 as they noticed a problem with the former pair of primers. Nevertheless, in Fig.4, the legend is not highlighted in red, being unclear whether the pair of primer that they used for this experiment may still correspond to the old one, as in Fig.S1 and S3. The authors should provide the sequence of the primers they used and confirm that they used the same set for all the expression profiles included in the paper.

Fig.S5g: As already requested by the reviewer, the authors should quantify the amount of UCP1 expressed by using another quantitative method (e.g western blot).

Text, line 120. The AdipoqCreERT2 transgenic mouse is not specific for generating WAT specific ko mice. The deletion is also present in BAT. The authors should correct that.

Text, line 201: UCP1 and Cidea are thermogenic markers, not only beige markers as well as pparg and Cebpb are adipogenic markers not only related to browning.

Answer to reviewers' comments

Reviewer #1 (Remarks to the Author):

This new version of the manuscript is stronger and the authors did a great job in the revision process.

We thank the reviewer for his/her nice comment regarding our revised manuscript.

Reviewer #2 (Remarks to the Author):

"We appreciate the effort of the authors in answering the concerns we raised. However, the main question remains open: which is the contribution/relevance of TGR5 dependent scWAT browning and mitochondrial biogenesis on the overall thermogenic phenotype of the mice. The authors in a previous paper and a review suggest already the importance of mechanisms involving TGR5 for BAT activation and its contribution to modulate energy expenditure (Watanabe et al, Nature, 2006; Houten et al, Embo J, 2006), so what is the contribution of TGR5 activation to BAT thermogenesis vs scWAT browning with respect to the mouse metabolic phenotype, when cold exposed or fed an HFD is important. Moreover, because the authors claim that TGR5 could represent a targetable candidate to prevent obesity and related metabolic complication. All in all, the data presented about the mechanism by which TGR5 activation induce mitochondrial biogenesis and browning is interesting but do not answer the main question.

We thank the reviewer for his/her comments, below we address his/her concerns regarding our revised manuscript.

The main purpose of this study was 1) to determine whether the bile acid-TGR5 axis can induce beiging and 2) to identify the molecular and cellular mechanisms behind this process. In the context of the reviewer's criticism, it is important to emphasize that we never claimed that the TGR5-dependent remodeling found in the scWAT is the only and unequivocal explanation for the thermogenic effect and for the overall phenotype described in this manuscript. We are indeed aware of the previously described mechanisms in the BAT as well as in other tissues, and we absolutely agree that it is the combination of TGR5-dependent effects in the different adipose depots that explains the overall beneficial thermogenic phenotype.

Currently, it remains challenging to distinguish effects from the BAT vs the scWAT when studying a beiging factor expressed in both fat depots. To our knowledge, there is no ideal Cre model available to appropriately address this question (for details see our first rebuttal). The only alternative would be to silence TGR5 by means of a shRNA lentiviral vector directly injected into the target fat depot (Balkow, A. et. al. *Journal of Biological Methods*, 2016; Ng, I. et. al. *Experimental and Molecular Medicine*, 2017; Gnad, T. et. al. *Nature*, 2014) and then repeat the interventions (e.g. cold exposure and HFD), associated with energy expenditure measurements. However, this approach has also technical limitations,

for instance the deletion of the receptor will not be 100%, and the possibility that we could still see activation in the presence of bile acids or INT-777 exist. In addition, we know that the interventions used in this work such as cold exposure and HFD, promote the renewing (cell proliferation and differentiation) of adipocytes that will not be targeted. Moreover, even considering the use of this approach (knowing in advance its limitations previously mentioned), it will take a lot of resources and time, and would definitely be beyond the scope of the current study.

Finally, we feel that this request is distracting from the essence of our work, which is the identification of mitochondrial fission as a critical process by which bile acids or TGR5 activators potentiate mitochondrial respiration. Within the context of beige fat remodeling, this mechanism is entirely novel and contributes to the general understanding of the molecular mechanisms involved in beiging. We are fully convinced that these findings are relevant to the field and will set the stage for new approaches to cure metabolic disorders.

Reply to reviewer's major comments

Fig.1a and 3a. The authors present the body weight gain of the mice, instead of the growth curve. Is it because the body weight of the mice were not similar between the experimental groups at the start of the experiment?

The body weight of the mice at the beginning of the experiment was similar between the groups. We decided to present the data as body weight gain since the intervention lasted only one week and we didn't have a real growth curve. We feel that it would not be appropriate to represent these data as a growth curve since we only have two measurements (before and after the one week intervention) and it is not a long-term experiment like the one we presented in Fig. 4a.

Fig 1a-3a. As mentioned in their answer to the reviewer comments, the authors designed a new set of oligonucleotides for Cidea and CD137 as they noticed a problem with the former pair of primers. Nevertheless, in Fig.4, the legend is not highlighted in red, being unclear whether the pair of primer that they used for this experiment may still correspond to the old one, as in Fig.S1 and S3. The authors should provide the sequence of the primers they used and confirm that they used the same set for all the expression profiles included in the paper.

As mentioned by this reviewer, we forgot to highlight in red the legends corresponding to the change of primers in all the figures where they were used. However, we repeated the measurements of Cidea and CD137 expression using the new pair of primers designed in all the expression profiles included in the paper. The sequences are the following:

Used in all mice cohorts, primary adipocytes and 3T3-L1 cell line:

CATGATCTTGGAAAAGGGACAG Cidea mouse FW

ATCGTGGCTTTGACATTGAGAC Cidea mouse RV

CCTTGCAGGTCCTTACCTTGT Cd137 mouse FW
GTTGCTTGAATATGTGGGGGA Cd137 mouse RV

Used in SGBS human cells:

TTATGGGATCACAGACTAAGCGA Cidea human FW
TGCTCCTGTCATGGTTGGAGA Cidea human RV

AGCTGTTACAACATAGTAGCCAC Cd137 human FW
TCCTGCAATGATCTTGTCTCT Cd137 human RV

Fig.S5g: As already requested by the reviewer, the authors should quantify the amount of UCP1 expressed by using another quantitative method (e.g western blot).

According to the previous concern from the reviewer: “**Fig.S4c The authors should provide a quantification of the UCP1 expression assessed by IF**”, we understood that the reviewer was only asking for the quantification of the immunofluorescence (IF) presented here below (old Fig. S4c) (Actually, Western blot analysis was not specified by this reviewer in the first version).

Supplementary Figure 4

The quantification of the IF clearly shows the upregulation of UCP1 protein in differentiated adipocytes after Tgr5 activation by the synthetic agonist INT-777 and the secondary bile acid LCA compared to the control (New Supplementary Fig. 5C and D, here below).

Supplementary Figure 5

Text, line 120. The AdipoqCreERT2 transgenic mouse is not specific for generating WAT specific ko mice. The deletion is also present in BAT. The authors should correct that.

We thank the reviewer for this observation, we will modify line 120: “we generated an inducible WAT-specific *Tgr5*^{-/-} model . . .” as follows: “we generated an inducible adipose tissue-specific *Tgr5*^{-/-} model...”, to avoid misunderstanding.

Text, line 201: UCP1 and Cidea are thermogenic markers, not only beige markers as well as pparg and Cebpb are adipogenic markers not only related to browning. "

We agree with this reviewer that the genes above have functions beyond the current description. However, since these transcripts are usually measured in the context of beige remodeling and to facilitate reading fluency in the manuscript, we preferred to use this terminology. If needed, we can specify the functions of these genes in the manuscript, as was done in the first version.